# OsSRF8 interacts with OsINP1 and OsDAF1 to regulate pollen aperture formation in rice

Keyi Chen [1,4], Qiming Wang[1,4], Xiaowen Yu[1], Chaolong Wang[1], Junwen Gao[1], Shihao Zhang[1], Siqi Cheng[1], Shimin You [1], Hai Zheng[1], Jiayu Lu[1], Xufei Zhu[1], Dekun Lei[1], Anqi Jian[1], Xiaodong He[1], Hao Yu[1], Yun Chen[1], Mingli Zhou[1], Kai Li[1], Ling He[1], Yunlu Tian[1], Xi Liu[1], Shijia Liu[1], Ling Jiang[1], Yiqun Bao[2], Haiyang Wang [3], Zhigang Zhao [1] ✉ & Jianmin Wan[1,3] ✉

In higher plants, mature male gametophytes have distinct apertures. After pollination, pollen grains germinate, and a pollen tube grows from the aperture to deliver sperm cells to the embryo sac, completing fertilization. In rice, the pollen aperture has a single-pore structure with a collar-like annulus and a plug-like operculum. A crucial step in aperture development is the formation of aperture plasma membrane protrusion (APMP) at the distal polar region of the microspore during the late tetrad stage. Previous studies identified OsINP1 and OsDAF1 as essential regulators of APMP and pollen aperture formation in rice, but their precise molecular mechanisms remain unclear. We demonstrate that the Poaceae-specific *OsSRF8* gene, encoding a STRUBBELIG-receptor family 8 protein, is essential for pollen aperture formation in *Oryza sativa*. Mutants lacking functional *OsSRF8* exhibit defects in APMP and pollen aperture formation, like loss-of-function *OsINP1* mutants. *OsSRF8* is specifically expressed during early anther development and initially diffusely distributed in the microsporocytes. At the tetrad stage, OsSRF8 is recruited by OsINP1 to the pre-aperture region through direct protein-protein interaction, promoting APMP formation. The OsSRF8-OsINP1 complex then recruits OsDAF1 to the APMP site to co-regulate annulus formation. Our findings provide insights into the mechanisms controlling pollen aperture formation in cereal species.

In higher plants, successful pollination and fertilization are crucial for the success of reproduction. The male gametes, sperm cells are non-motile and thus require transportation to the ovules via the pollen tube to execute successful double fertilization[1–4]. After pollination, a pollen grain germinates and pollen tube grows from pollen aperture delivering two sperm cells to the embryo sac to fuse with the egg cell and the central cell, respectively, to complete the double fertilization[5,6]. During the entire pollination and fertilization process, pollen aperture not only serves as the channel for pollen tube

germination and growth, but also provides an important site for water uptake during pollen rehydration[7–10]. The pollen aperture formed by the gaps in exine deposition, allowing these sites to serve as an outlet for pollen tube germination and pollen hydration. As a species-specific characteristic, the pollen aperture exhibits large variations in number (from zero to many), morphology, position, and orientation, and so on[11–16]. The two most common pollen aperture patterns are a single-pore present in pollen grains of many monocots, and three apertures found in the majority of edicots[7,17–20]. However,

[1]State Key Laboratory for Crop Genetics & Germplasm Enhancement and Utilization, Nanjing Agricultural University, Zhongshan Biological Breeding Laboratory, Nanjing 210095, China. [2]School of Life Sciences, Nanjing Agricultural University, Nanjing 210095, China. [3]National Key Facility for Crop Gene Resources and Genetic Improvement, Institute of Crop Science, Chinese Academy of Agricultural Sciences, Beijing 100081, China. [4]These authors contributed equally: Keyi Chen, Qiming Wang. ✉e-mail: zhaozg@njau.edu.cn; wanjianmin@caas.cn

the molecular mechanisms regulating pollen aperture formation remain largely unclear.

Pollen aperture in rice is a single-pore structure located at the distal polar region, consisting of a slightly raised collar-like annulus and a plug-like operculum (Supplementary Fig. 1a). Based on the cross section of rice pollen aperture, the area of aperture lacks exine deposition, and the fibrillar-granular layer and Zwischenkörper layer that distributed under the operculum are joined with the annulus, so that a closed compartment forms within the pollen grain (Supplementary Fig. 1b). Like exine formation, the first sign of aperture in rice is readily visible at the late tetrad stage, indicating that the aperture pattern is determined during microsporogenesis[7,20,21]. During the time, the plasma membrane (PM) is polarized to form the aperture plasma membrane protrusion (APMP), which prevents deposition of primexine/exine at these regions, a critical step in the aperture pattern formation[19,22–24]. Then, the operculum and annulus begin to develop in the free microspore stage (S9), and their compositions are mainly sporopollenin, which is highly resistant lipid-rich complexes derived from the tapetum[25–27]. The fibrillar-granular layer and Zwischenkörper layer comprised of well-developed callose/pectin substances are formed in the vacuolated microspore stage (S10) and binucleate pollen stage (S11), respectively[27–30]. Based on these characteristics, the formation of rice pollen aperture can be roughly divided into two consecutive processes: PM polarization and the formation of APMP defining the aperture pattern, and decoration of the aperture including formation of the annulus and operculum (Supplementary Fig. 2).

Previous forward genetic screening had identified several regulators of aperture formation in rice[7]. Among them, *OsINP1* (*INAPERTURATE POLLEN1*), orthologous to a key aperture factor *AtINP1* in Arabidopsis, controls the formation of the APMP on rice pollen. It encodes a protein of unknown biochemical function with a single recognizable DELAYED IN GERMINATION1 (DOG1) domain[7,20]. Dysfunction of *OsINP1* leads to the absence of aperture on the pollen surface and precludes pollen tube germination. *OsDAF1* (*DEFECTIVE IN APERTURE FORMATION1*), encoding a legume-lectin receptor-like kinase, is essential for the formation of the annulus and thus for male fertility. OsINP1 and OsDAF1 proteins are initially distributed in microspore mother cells (MMCs), and they sequentially and specifically aggregate into a tiny ring that corresponds the pre-aperture region at the late tetrad stage. When the tetrad microspores are separated, OsINP1 is distributed below the gap (lacking primexine/exine), and OsDAF1 is located close to the edge of the annulus. OsDAF1 functions downstream of OsINP1, and OsINP1 acts to regulate polar distribution of OsDAF1 at the future aperture sites through direct interaction with the cytoplasmic domain of OsDAF1. Despite these progresses made in this area, the detailed molecular mechanisms of OsINP1 and OsDAF1 regulating APMP and annulus formation still remain unclear.

In this study, we report the isolation and identification of the gene *STRUBBELIG-receptor family 8* (*OsSRF8*), which encodes a predicted receptor kinase required for pollen aperture formation in rice. We demonstrate that OsINP1 recruits and co-locates with OsSRF8 to jointly control the formation of APMP. Furthermore, the OsINP1-OsSRF8 protein complex recruits OsDAF1 to regulate the annulus formation. Our results elucidate a critical regulatory mechanism controlling APMP and pollen aperture formation in rice.

## Results

### The absence of aperture in the *Ossrf8* mutant leads to male sterility
To look for novel regulators of pollen aperture formation in rice, we screened for mutants defective in male fertility and identified a rice mutant (named *Ossrf8*, see below) that lacks pollen aperture and is completely male sterile. At the seedling and heading stages, the *Ossrf8* mutant plants showed no obvious differences from the wild type (WT).

However, at maturity, the *Ossrf8* mutant was completely sterile (Supplementary Fig. 3a, b). No differences in anther development and dehiscence, and pollen grain starch filling were observed between the WT and the *Ossrf8* mutant (Supplementary Fig. 3c–q). Besides, mature embryo sac of the *Ossrf8* mutant was normal as the WT (Supplementary Fig. 3r, s). In vitro and in vivo pollen germination experiments showed that there is no pollen tube growth in the *Ossrf8* mutants (Fig. 1a, b and Supplementary Fig. 4). These data suggest that pollen lesion might be responsible for male sterility in the *Ossrf8* mutant.

Scanning electron microscopy (SEM) observation revealed no pollen aperture in the *Ossrf8* mutant, which is consistent with the failure of pollen germination (Fig. 1c, d). Subsequently, transmission electron microscopy (TEM) showed no significant differences between WT and mutant pollen (including pollen exine and intine) except for pollen aperture (Fig. 1e–h). According to previous studies, APMP generated at the pre-aperture sites is critical for aperture formation[7,22]. To explore whether the formation of APMP is affected in *Ossrf8* mutants, we observed the state of PM in WT and *Ossrf8* tetrad using the cell mask deep red staining experiment. The results showed that there was distinct APMP in the four corners of the WT tetrads, but no clear APMP was visible in the *Ossrf8* mutant (Fig. 1i–l). Taken together, these results indicate that the *Ossrf8* mutation causes the absence of APMP and disrupts formation of pollen aperture, resulting in male sterility in the *Ossrf8* mutant.

### *OsSRF8* encoding a Poaceae-specific receptor kinase regulates pollen aperture formation
To identify the molecular mechanism underlining the *Ossrf8* mutant phenotype, a map-based cloning approach was adopted. In an $F_2$ population of *Ossrf8* mutant and N22 (*Oryza sativa ssp. indica*), fertile and sterile plants segregated in a 3:1 ratio (1130 fertile versus 420 sterile, $\chi^2 = 3.63 < \chi^2_{0.05} = 3.84$; $P > 0.05$), indicating that the *Ossrf8* defect was caused by a recessive mutation in a single nuclear gene (Fig. 2a). The *OsSRF8* locus was mapped to a region between the molecular markers, DY-8 and I2-3, on chromosome 2, and subsequently fine-mapped to a 150 kb fragment between markers DY9 and DY15. Seventeen predicted open reading frames were found in this region. To narrow down the list of candidate genes, we performed next-generation sequencing on pooled *Ossrf8* mutant and WT genomic DNA, respectively. A single nucleotide polymorphism was identified in *LOC_Os02g10110*, which corresponds the *STRUBBELIG-receptor family 8* (*SRF8*) gene encoding a putative leucine-rich repeat receptor-like kinase (LRR-RLK)[31]. The mutation (G-A) resulted in a premature translation termination at codon 377 (Fig. 2a). Thus, *LOC_Os02g10110* was considered as the candidate gene for *OsSRF8*.

To verify this, we conducted a genomic complementation assay. Wild-type genomic sequences including the entire coding region (with or without eYFP) and 2.24-kb promoter of *LOC_Os02g10110* was introduced into the *Ossrf8* homozygous plants, and pollen fertility was restored, verifying that *LOC_Os02g10110* is indeed the gene responsible for the aperture defects in the *Ossrf8* mutant (Fig. 2b–d, g–i, l–n, Supplementary Fig. 5 and Supplemental Table 1). Consistently, two knock-out mutant lines (*Cr-Ossrf8-1* and *Cr-Ossrf8-2*) of *LOC_Os02g10110* generated using the CRISPR-Cas9 technology also lacked APMP and pollen aperture and exhibited complete sterility (Fig. 2e, f, j, k, o–q, Supplementary Fig. 5 and Supplemental Table 2). Taken together, our results demonstrate that *LOC_Os02g10110* is the gene whose disruption is responsible for the mutant phenotype and that the 2.24-kb upstream region is sufficient to drive proper expression of *OsSRF8*.

Sequencing analysis of the cDNA sequence showed that *OsSRF8* consists of sixteen exons and fifteen introns (Fig. 2a). The predicted OsSRF8 protein consists of 757 amino acids, with a predicted signal peptide (1-25), multiple LRR repeat units (26-320), one predicted transmembrane domain (321-343) and a protein kinase domain (344-

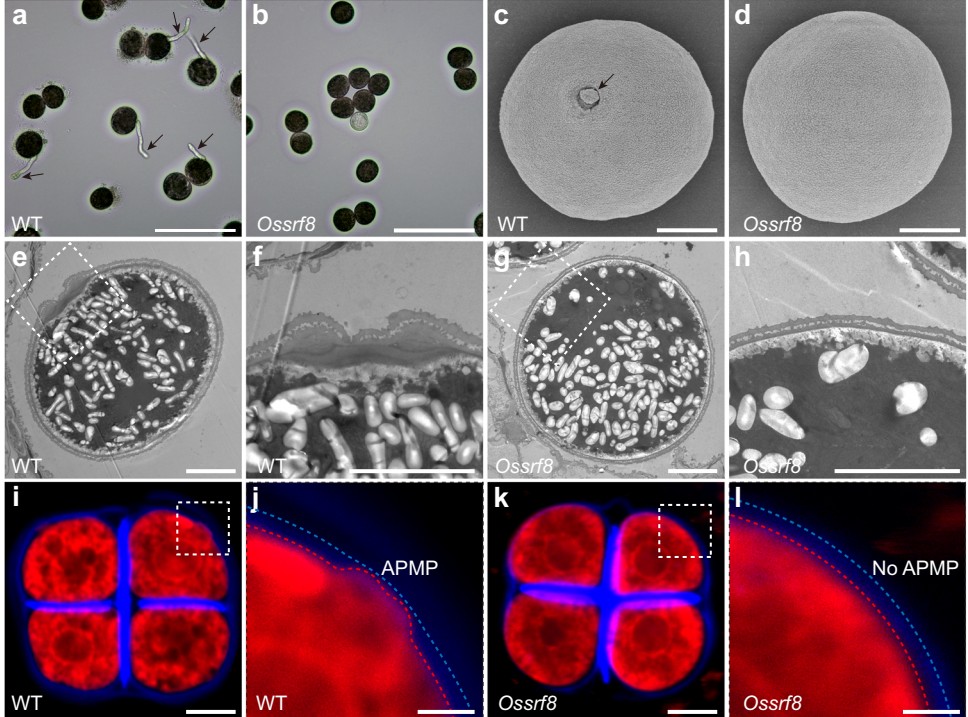

**Fig. 1 | Phenotype comparison between the WT and *Ossrf8* mutant. a, b** In vitro pollen germination assays of WT and the *Ossrf8*. The black arrows indicate pollen tubes. Phenotypes in (**a, b**) were observed at least three times independently with similar results. Scale bars, 100 μm. **c, d,** SEM micrographs show that there is no aperture on the *Ossrf8* pollen. The black arrow indicates the aperture. Three plants (≥30 pollen grains per plant) were imaged in (**c, d**) with similar results. Scale bars, 10 μm. **e–h** TEM images of mature pollen. **f** and (**h**) represent enlarged views of the dashed square, respectively. ≥10 pollen grains were imaged in (**e, g**) with similar results. Scale bars, 10 μm. **i–l** Confocal images of tetrads stained by Calcofluor White (blue, callose wall) and Cell Mask Deep Red (red, membranous structures). APMP (dashed square) can be observed in wild-type tetrads, but not in mutant tetrads. **j** and **l** represent enlarged views of the dashed square, respectively. The red dotted lines indicate the boundary of the plasma membrane and the blue dotted lines indicate the outer boundary of the callose wall of microspore at the tetrad stage. Experiments in (**l, k**) were repeated three times with similar results. Scale bars, 10 μm in (**i, k**), 2 μm in (**j, l**).

757) (Supplementary Fig. 6). A phylogenetic relationship analysis showed that OsSRF8 was relatively conserved in Poaceae (above 67% amino acid identity), and only one copy existed in the rice genome (Supplementary Fig. 7). In addition to Poaceae, some other species also contained homologous proteins, but these proteins shared lower identity with OsSRF8. This observation suggests that OsSRF8 and its orthologues likely play a conserved role in regulating pollen aperture formation in Poaceae.

## OsSRF8 specifically accumulates at the pre-aperture region at the tetrad stage

Analysis of the publicly available gene expression data from the RiceXPro public databases (https://ricexpro.dna.affrc.go.jp) showed that the *OsSRF8* gene is specifically expressed during the early stages of anther development when the aperture begins to form (Supplementary Fig. 8). In order to further determine whether *OsSRF8* is expressed during the aperture formation, we introduced an expression cassette containing the nuclear marker histone H2B tagged with enhanced green fluorescent protein (EGFP) under the control of the *OsSRF8* promoter (*pOsSRF8:H2B-EGFP*) into the WT plants (Supplementary Table 3). The nuclear EGFP signal was firstly observed in the MMCs, dyad-stage microspores, tetrad-stage microspores, and early-stage young free microspores (Supplementary Fig. 9a–e). However, upon investigation of the microspore fluorescence signal at the vacuolated microspore stage, it was observed that only 14 out of 57 microspores retained fluorescence, suggesting that the signal started to disappear at the vacuolated microspore stage (Supplementary Fig. 9f and Supplementary Table 4). These results are consistent with a critical role of OsSRF8 in pollen aperture formation. To determine

the subcellular localization of the OsSRF8 protein, we transiently expressed an EGFP-tagged OsSRF8 fusion protein in rice protoplasts and observed co-localization of the EGFP fluorescent signal with the PM marker (Supplementary Fig. 10a–h). Further, plasmolysis experiments verified PM localization of the OsSRF8 protein in tobacco epidermal cells expressing EGFP-tagged OsSRF8 fusion protein (Supplementary Fig. 10i–w). Hence, we concluded that OsSRF8 is localized on the PM.

To further probe the role of OsSRF8 in pollen aperture formation, we generated *pOsSRF8:gOsSRF8-eYFP* transgenic plants in the *Ossrf8* mutant background. Pollen aperture was restored in all positive transgenic plants, indicating that the transgene is functional (Fig. 2d, i, n and Supplementary Table 5). Confocal microscope examination revealed that the OsSRF8-eYFP fluorescence signals were randomly distributed in the cytosol and on the PM at the MMC stage (Fig. 3a). After meiosis I and meiosis II, the OsSRF8-eYFP signal remained diffuse in the cytosol at the dyads and early tetrads (Fig. 3b, c). After further development of the tetrads, the OsSRF8-eYFP signal started to aggregate at the four distal corners of the tetrads, although faint YFP signal was still observed in the cytosol and on the rest part of the PM (Fig. 3d). At the late tetrad stage, the OsSRF8-eYFP signal was further aggregated at the four corners of the tetrads, with no visible fluorescent signal left in the cytosol (Fig. 3e). When the microspores were dissociated from the tetrads, the OsSRF8-eYFP signal was clustered in a ring-like structure corresponding to the future aperture site (Fig. 3f). When the pollen aperture was essentially formed at the vacuolated microspore stage, the OsSRF8-eYFP signal was sandwiched between the annulus and operculum (Fig. 3g, h). Thus, the localization of OsSRF8 is consistent with its critical role in pollen aperture formation.

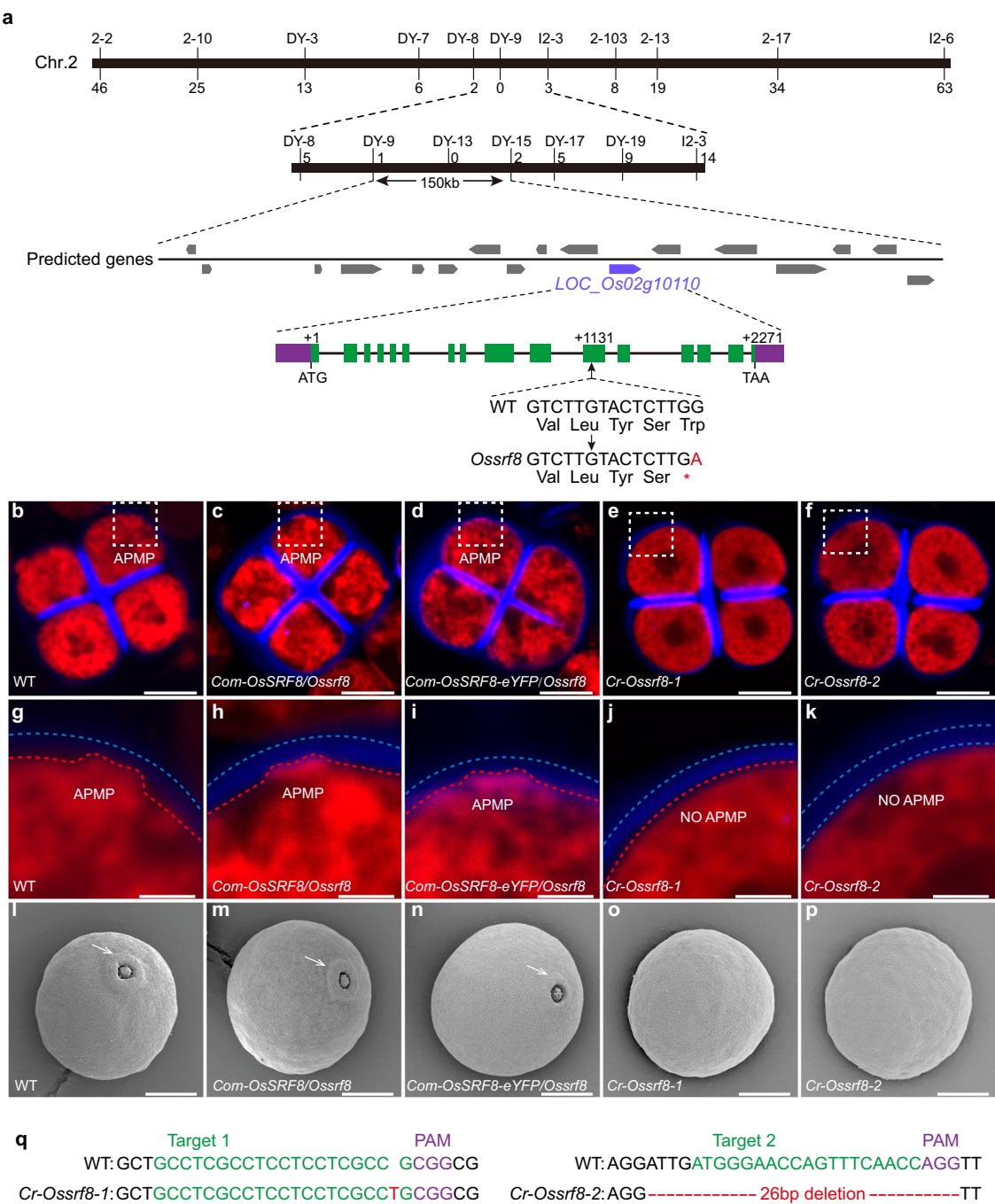

**Fig. 2 | Cloning and identification of *OsSRF8*. a** Fine-mapping of *OsSRF8*. The molecular markers and numbers of recombinants are noted. *OsSRF8* is located on chromosome 2 with seventeen predicted ORFs in the interval between the markers DY-9 and DY-15. *LOC_Os02g10110* is identified as the candidate gene. Mutant *Ossrf8* harbors (G-A) single substitution results in premature translation termination. ATG, start codon; TAA, stop codon. Confocal images of tetrads of WT (**b**), the *Ossrf8* mutant rescued by genomic *OsSRF8* sequence or *OsSRF8* fused with eYFP (**c**, **d**), *Cr-Ossrf8-1* (**e**) and *Cr-Ossrf8-2* (**f**). APMP is marked with the dashed square. Experiments in (**b–f**) were repeated three times with similar results. Scale bars, 10 μm. **g–k** The images of the enlarged APMP regions corresponding to the dashed areas in a-e, respectively. The red dotted lines indicate the boundary of the plasma membrane and the blue dotted lines indicate the outer boundary of the callose wall of microspore at the tetrad stage. Scale bars, 3 μm. Scanning electron microscopy observation of WT (**l**), the *Ossrf8* mutant rescued by genomic *OsSRF8* sequence or *OsSRF8* fused with eYFP (**m**, **n**), *Cr-Ossrf8-1* (**o**) and *Cr-Ossrf8-2* (**p**) mature pollen grains. ≥30 pollen grains were imaged in (**l–p**) with similar results. Scale bars, 10 μm. **q** The display of target sites of CRISPR/Cas9. At the target site 1, *Cr-Ossrf8-1* sequences exhibit a T (red) base insertion mutation; at the target site 2, *Cr-Ossrf8-2* sequences have a 26-bp (red) deletion mutation. Green indicates the 20-bp CRISPR-Cas9 target sequence.

## The OsINP1-OsSRF8 protein complex regulates the formation of APMP

Previous studies have revealed that OsINP1, encoding a protein of unknown biochemical function, plays an essential role in rice aperture formation. Disruption of OsINP1 results in the absence of APMP and pollen aperture, and complete male sterility[7,9,20,22]. Thus, we tested whether OsSRF8 and OsINP1 physically interact with each other. OsSRF8 was divided into the extracellular (OsSRF8(N), 26-320) and intracellular segments (OsSRF8(C), 344-757) in a yeast two-hybrid assay. The results showed that only OsSRF8(N) interacted with OsINP1

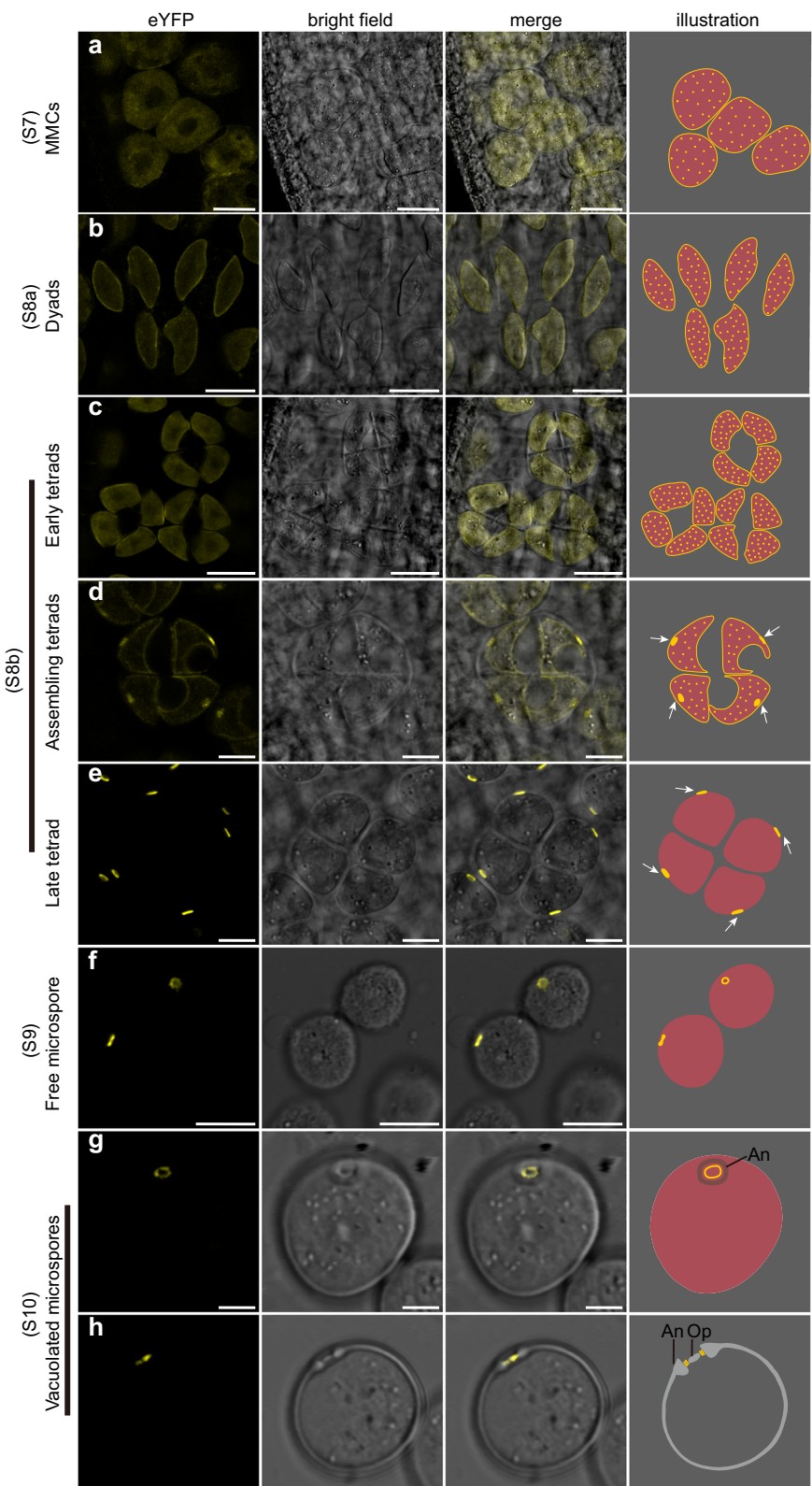

**Fig. 3 | OsSRF8 accumulates polarly at the site of future aperture formation.** Confocal imaging and illustrations of microspores expressing *pOsSRF8:gOsSRF8-eYFP* at S7 (**a**), S8a (**b**), S8b (**c**–**e**), S9 (**f**) and S10 (**g**, **h**). White arrows indicate the future aperture sites on the tetrad. An annulus, Op operculum. Five independent $T_0$ and $T_1$ lines were imaged, with similar results. Scale bars, 20 μm in (**a**-**c**), 10 μm in (**d**–**f**) and 5 μm in (**g**, **h**).

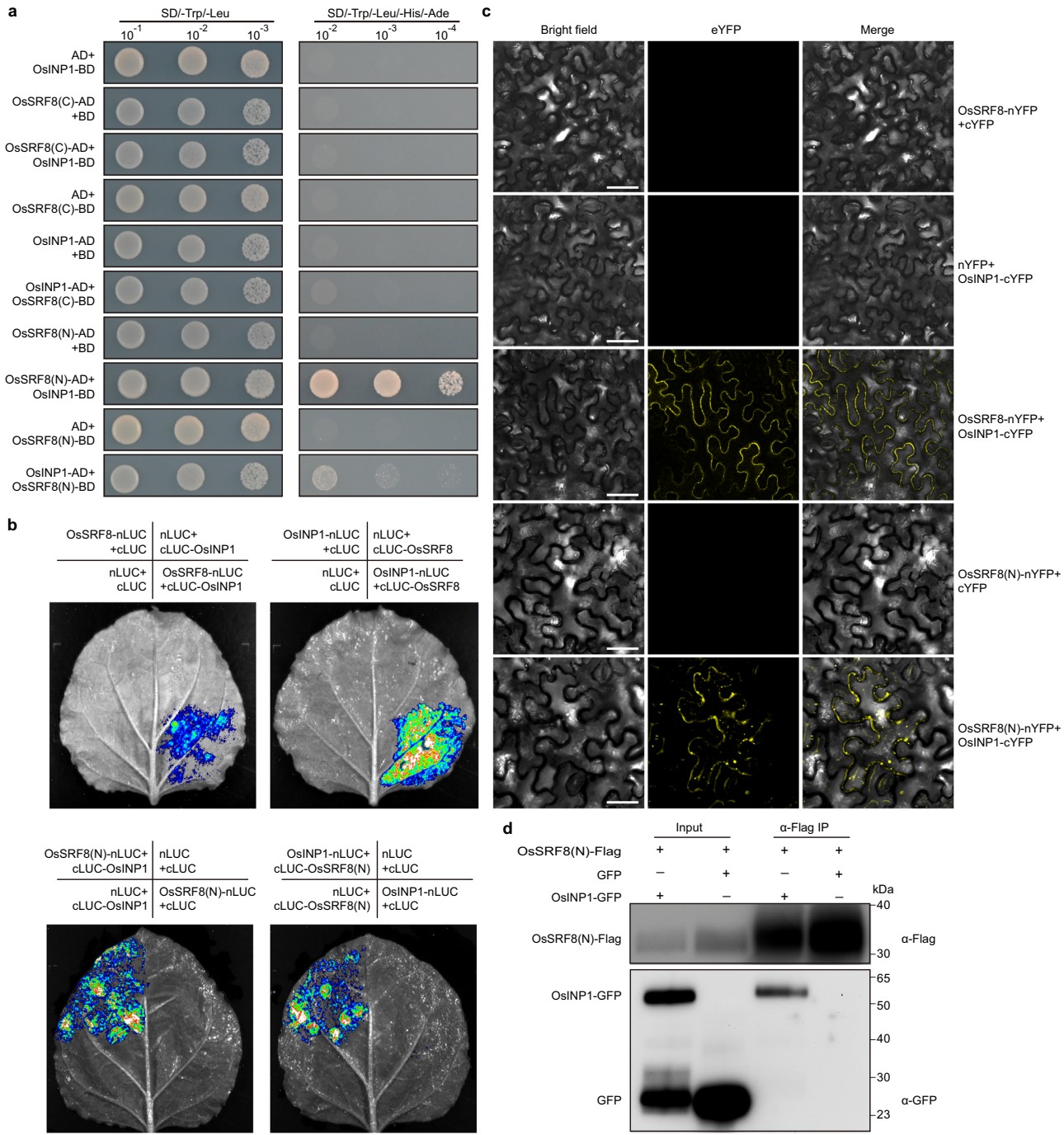

**Fig. 4 | OsSRF8 interacts with OsINP1. a** Yeast two-hybrid (Y2H) assay verified the interaction of OsSRF8 and OsINP1. OsSRF8 (N) and OsSRF8 (C) represent the N-terminal domain (amino acids 26-320) and C-terminal domain (amino acids 344-757) of the OsSRF8 protein, respectively. AD activating domain, BD DNA-binding domain, SD synthetic defined medium. **b** Split-luciferase assay testing the interaction between OsSRF8 and OsINP1 in the tobacco (*Nicotiana benthamiana*) leaves.

**c** Interaction between OsINP1 and OsSRF8 detected using Bimolecular fluorescence complementation (BiFC) assay. OsINP1 and OsSRF8 respectively were fused to the C- and N-terminal parts of YFP and co-transformed into tobacco leaves. Scale bars, 50 μm. **d** Co-immunoprecipitation experiments in the tobacco leaves. α-Flag IP indicates immunoprecipitation with anti-Flag antibody beads. All experiments were repeated three times, with similar results.

protein (Fig. 4a). Subsequently, we used the full-length OsSRF8 and the truncated OsSRF8(N) protein to perform interaction experiments with OsINP1. The results from both the luciferase complementation imaging (Luc) (Fig. 4b) and bifluorescent molecular complementation (BiFC) (Fig. 4c) showed that both the full-length OsSRF8 and the truncated OsSRF8(N) protein could interact with OsINP1. In addition, the truncated OsSRF8(N) protein could interact with OsINP1 protein in co-immunoprecipitation assay (Fig. 4d). These results suggest that

OsSRF8 forms a protein complex with OsINP1 to regulate pollen aperture formation in rice.

To substantiate the above notion genetically, we generated single (*Osinp1*) and double (*Ossrf8/Osinp1*) mutants through a combination of CRISPR/Cas9 technology and genetic crosses (Supplementary Tables 2 and 6). The *Osinp1* mutant contained a single base deletion (T) that resulted in altered-frame translation, the absence of APMP, and pollen grains completely lack aperture (Fig. 5a, b and Supplementary

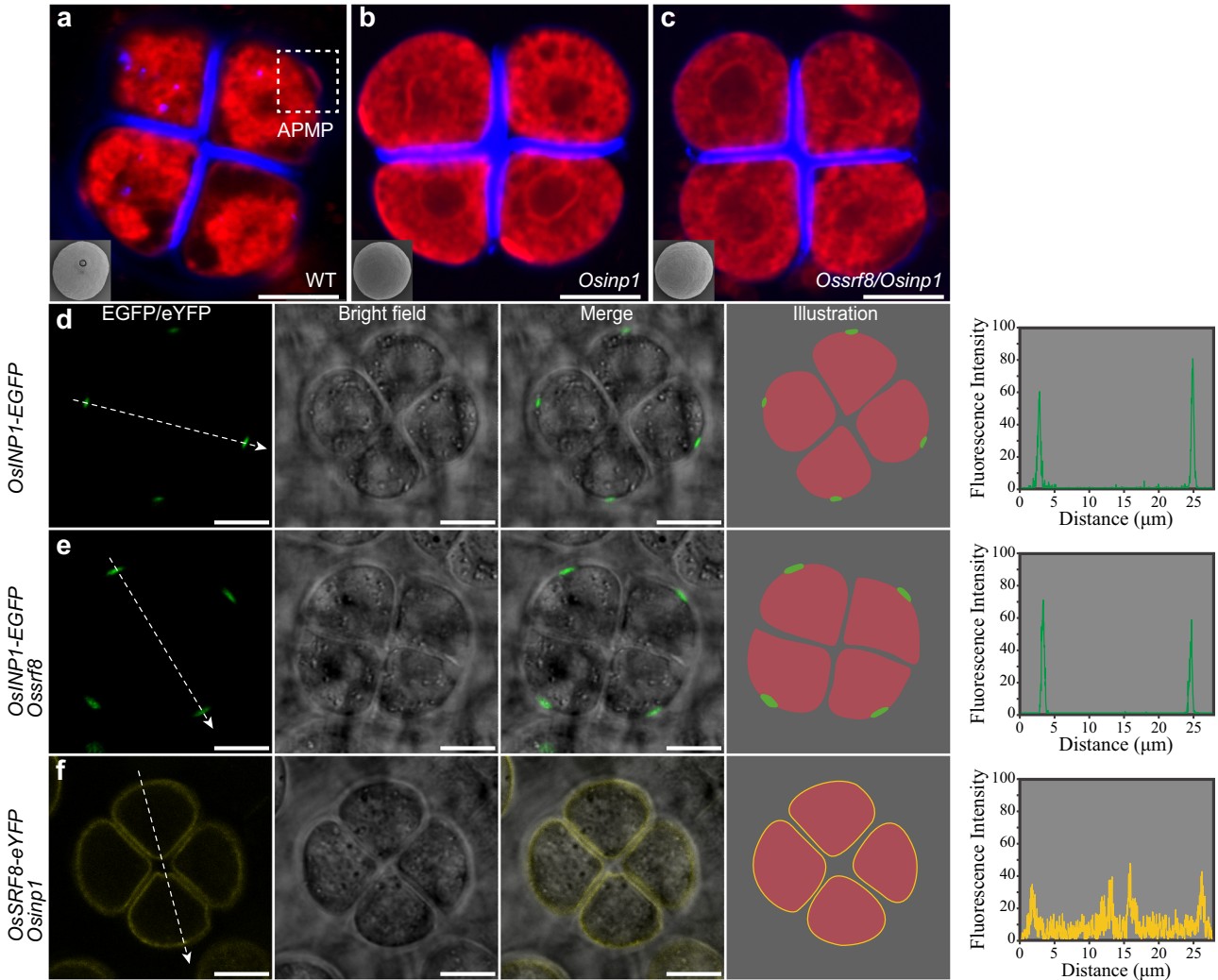

**Fig. 5 | Analysis of the genetic relationship between *OsSRF8* and *OsINP1.** The observation of tetrads and mature pollen of WT (**a**), *Osinp1* single (**b**), and *Ossrf8/ Osinp1* double mutant (**c**). The dashed square indicates APMP. Experiments were repeated three times, with similar results. Scale bars, 10 μm. Confocal images and illustrations of OsINP1-EGFP signal in tetrads of WT (**d**) and *Ossrf8* mutant (**e**) background. Scale bars, 10 μm. **f** Confocal optical sections and schematic diagrams of tetrads expressing OsSRF8-eYFP protein in the *Osinp1* background. Scale bars, 10 μm. Five independent T₁ lines were imaged, with similar results. The graphs on the right represent the fluorescence values after quantification and the white dashed arrows indicate the direction of fluorescence measurement.

Fig. 11m). The identical phenotypes observed in the single mutants of *Osinp1* and *Ossrf8* suggest that they operate within the same genetic pathway to regulate aperture pattern in pollen surface. To further explore potential synergistic interactions between *OsSRF8* and *OsINP1*, we generated the *Ossrf8/Osinp1* double mutant. As expected, the phenotype of the double mutant was identical to that of the single mutant (Fig. 5c and Supplementary Fig. 11a–f). In order to further explore the regulatory relationship between OsSRF8 and OsINP1, we transformed *pOsINP1:gOsINP1-EGFP* into the *Ossrf8* mutant background, meanwhile, transgenic plants expressing *pOsSRF8:gOsSRF8-eYFP* were crossed with *Osinp1* mutants (Supplementary Tables 7 and 8). We found that the localization pattern of OsINP1-EGFP was not affected in the *Ossrf8* mutant background (Fig. 5d, e). Notably, in *Osinp1* mutant, the polar location of OsSRF8-eYFP was completely disrupted and exhibited a random distribution on the PM (Fig. 5f). These results indicate that OsINP1 is required for proper targeting of the OsSRF8 protein to the pre-aperture region.

Previous reports have shown that OsINP1-eYFP is involved in the modification of the PM at the future aperture site, narrowing the distance between the PM and the callose wall, thereby preventing the deposition of sporopollenin at the future aperture sites[7]. In order to explore whether OsSRF8 has a similar localization to OsINP1 in the APMP, staining experiment of tetrads expressing the OsSRF8-eYFP fusion protein was performed (Calcofluor White, which stains callose wall and Cell Mask Deep Red, which stains PM). Distinct eYFP fluorescence signals were observed between the callose wall and the PM and these signals coincided with APMP (Fig. 6a). Furthermore, we introduced the vector co-expressing OsSRF8-mCherry and OsINP1-EGFP in the WT plants. we found that during the tetrad stage, the fluorescence signals of OsSRF8-mCherry and OsINP1-EGFP were colocalized (Fig. 6b). These observations collectively suggest that OsINP1 recruits OsSRF8 to APMP and co-regulate the formation of APMP.

## OsSRF8 interacts with OsDAF1 to regulate annulus formation

Previous studies reported that OsDAF1 is co-localized with OsINP1 and they interact directly to co-regulate annulus formation[7]. To investigate the regulatory relationship between OsSRF8 and OsDAF1, we first tested whether OsSRF8 could physically interact with OsDAF1. A series of interaction assays showed that OsSRF8 indeed interacted with OsDAF1 and that the C-terminal domain of OsSRF8 is responsible for interacting with the C-terminal domain of OsDAF1 (Fig. 7a–d). Thus, we speculated that the OsINP1-OsSRF8 complex formed at APMP may act to recruit

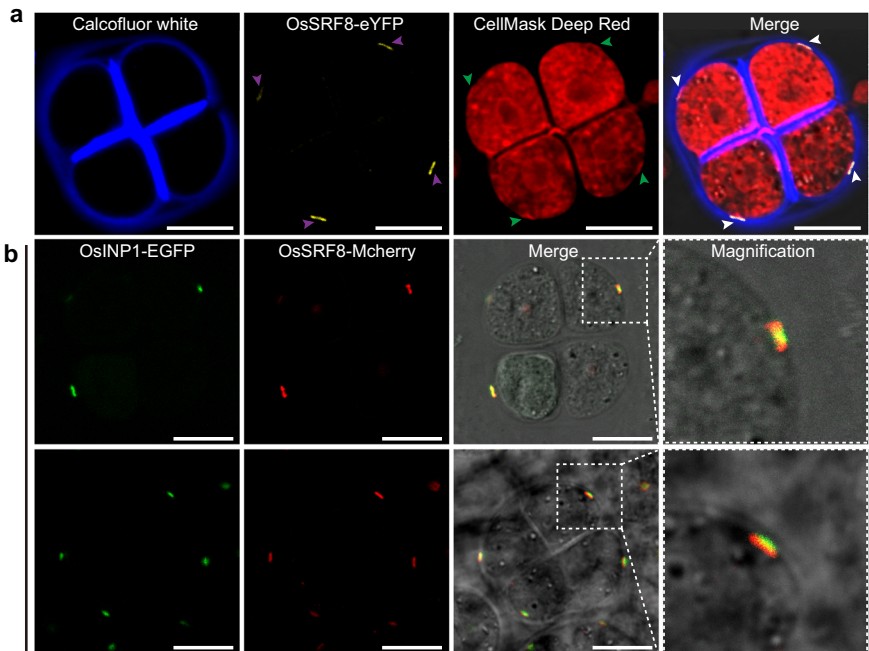

**Fig. 6 | OsSRF8 and OsINP1 are co-localized at the APMP. a** Tetrads of OsSRF8-eYFP transgenic plants are stained by Calcofluor White and Cell Mask Deep Red. The purple arrowheads represent the OsSRF8-eYFP localization, the green arrowheads represent the APMP, and the white arrowheads represent that OsSRF8 and APMP coincide with each other. **b** Colocalization of OsSRF8-mCherry and OsINP1-EGFP was observed from different angles. The area marked by a dashed line with square is shown by 6.25× magnification. All experiments were repeated at least three times, with similar results. Scale bars, 10 μm.

OsDAF1 to this area through physical interaction to promote subsequent formation of the annulus. To test this, we obtained an *Osdaf1* mutant through mutagenesis and generated *Ossrf8/Osdaf1* double mutants through genetic crosses (Supplementary Fig. 11g–l and Supplementary Table 9). At the same time, *Ossrf8/Osinp1/Osdaf1-1* triple mutants were obtained using the CRISPR/Cas9 technology to knock out *OsDAF1* in the background of heterozygous *Ossrf8/Osinp1* mutants (Supplementary Fig. 11g, k, l, n and Supplementary Table 10). Iodine-potassium iodide (I$_2$-KI) staining showed that most of the *Osdaf1* mutant pollen were aborted (Supplementary Fig. 12a, b, e, f). We speculated that the abortion of the *Osdaf1* mutant pollen is likely caused by leakage of starch granules due to the absence of the annulus. As expected both the *Ossrf8/Osdaf1* and *Ossrf8/Osinp1/Osdaf1-1* mutants showed an inaperturate pollen phenotype as the *Ossrf8* and *Osinp1* mutants. Interestingly, the starch-filling phenotype of the *Osdaf1* mutants could be restored by the *Ossrf8* and *Osinp1* mutations (Supplementary Fig. 12). Additionally, we obtained transgenic plants expressing the OsDAF1-eYFP in the *Ossrf8* mutant background, and found that the localization of the OsDAF1-eYFP protein was disrupted. In contrast to having concentrated fluorescent signals in the pre-aperture region in the WT background, most of the fluorescent signals exhibited a random distribution on the PM in the *Ossrf8* mutant background (Fig. 7e, f and Supplementary Table 11). On the contrary, the polar location of OsSRF8-eYFP was not affected in *Osdaf1* mutant background (Fig. 7g and Supplementary Table 12). These results suggest that the upstream OsINP1-OsSRF8 protein complex recruits OsDAF1, and three of them work together to regulate annulus formation.

## Discussion

The aperture generates specific patterns on the pollen surface and it tend to be highly conserved within a species[8–10]. Rice pollen aperture is a complex single-pore structure comprising annulus and operculum[7]. In the process of aperture formation, the generation of APMP is a critical step, indicating that the tetrad microspore cells have been polarized in advance. APMP defines the location and shape of aperture by preventing the deposition of primexine and sporopollenin there[7,22].

In this study, we demonstrate that the rice OsSRF8 promotes the formation of APMP on tetrads microspores, and that OsSRF8 is recruited to the pre-aperture region through physical interaction with OsINP1 to promote APMP formation. Then, the OsINP1-OsSRF8 protein complex recruits OsDAF1 to this region to regulate annulus formation. Combining our results and previous studies, we propose a speculative model for pollen aperture formation in rice (Fig. 8). Firstly, OsINP1 becomes localized at the pre-aperture region into a tiny ring at the late tetrad stage in response to some unknown polarity signal. Next, OsINP1 recruits OsSRF8 to this region through physical interaction, thereby promoting PM polarization and APMP formation through an unknown mechanism. Next, OsINP1 and OsSRF8 act together to recruit OsDAF1 to the APMP site to promote annulus formation. During S9 and S10, OsDAF1 is distributed near the edge of the annulus and may be involved in controlling the transport of sporopollenin to the annulus and condensing the Fibrillar-granular layer. However, mutation in *OsSRF8* causes absence of APMP and abnormal localization of OsDAF1, eventually forming pollen grains without aperture. Our results are consistent with the reported role of Arabidopsis INP1 in guiding the formation of APMP and three equidistant longitudinal apertures[20,22,23,32], and provide novel insights into the molecular mechanism of pollen aperture formation in rice.

Notably, here we show that both OsSRF8 and OsINP1 are required to form APMP and that targeting of OsSRF8 to the APMP region is dependent on OsINP1. Nevertheless, as the biochemical function of OsINP1 is still unknown, and thus, how it is targeted to the membrane of the pre-aperture region is an intriguing question. Presumably a position-dependent signal (s) on the microspore at the tetrad stage triggers precise targeting of OsINP1 to the pre-aperture region (distal polar region). The nature of the signal and how OsINP1 responds to the signal for protein targeting await further studies. Moreover, how the OsINP1-OsSRF8 protein complex promotes the PM to form a protruding structure is also unknown. Earlier studies suggest that lipid bilayer is a thin elastic sheet, fluid in plane but solid in bending, and the generation of PM protrusion can be understood as a balance of forces, which may arise from two pathways, one from the motor proteins and

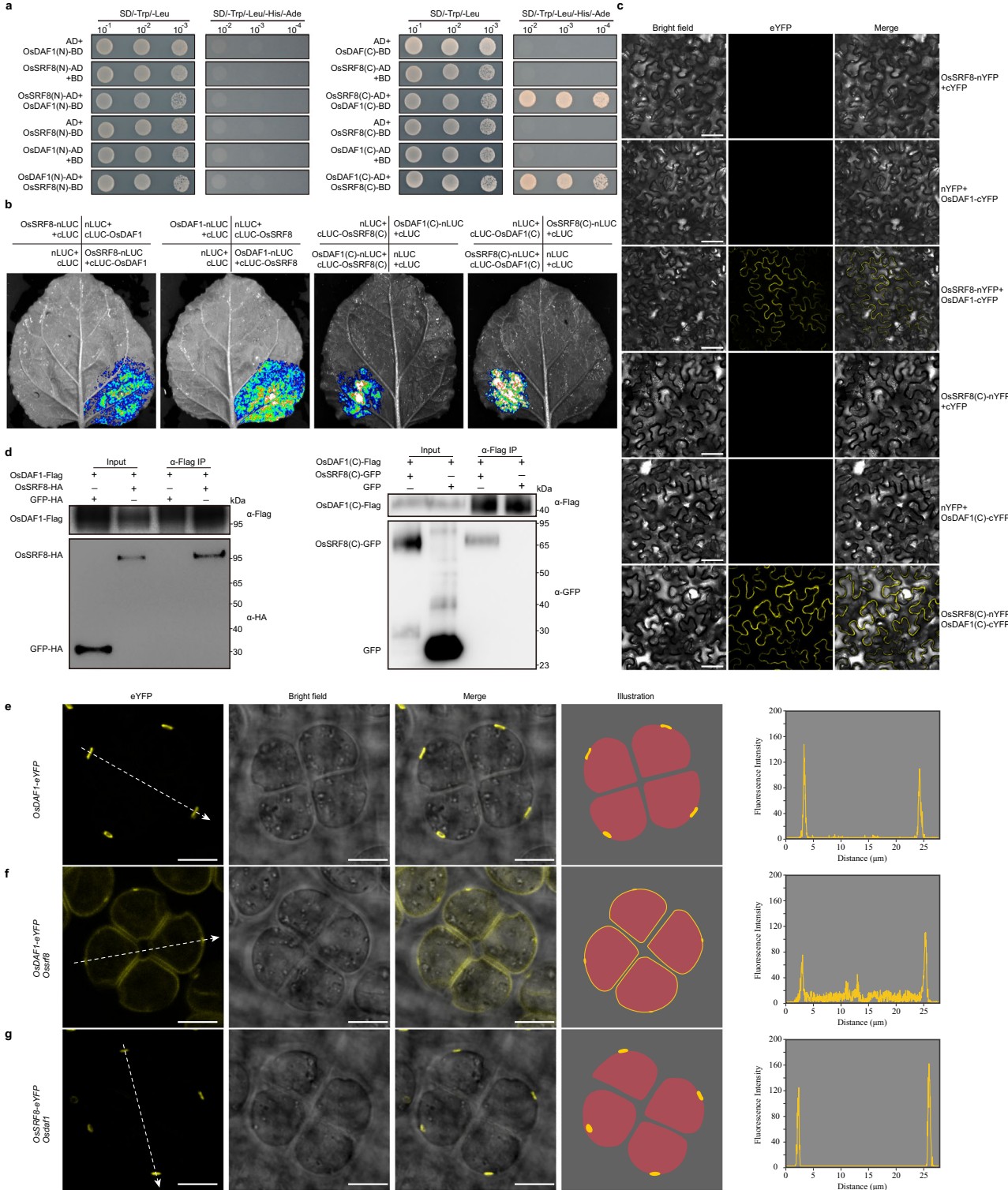

**Fig. 7 | OsSRF8 recruits OsDAF1 to regulate annulus formation. a** Y2H assay verifies the interaction of OsSRF8 and OsDAF1. OsDAF1 (N) and OsDAF1 (C) represent the N-terminal domain (amino acids 24-331) and C-terminal domain (amino acids 355-695) of the OsDAF1 protein, respectively. **b** Split-luciferase assay testing the interaction between OsSRF8 and OsDAF1 in the tobacco (*Nicotiana benthamiana*) leaves. **c** Interactions between OsSRF8 and OsDAF1 detected using BiFC assay. OsSRF8 and OsDAF1respectively were fused to the N- and C-terminal parts of YFP and co-transformed into tobacco leaves. Scale bars, 50 μm. **d** Co-immunoprecipitation experiments in the rice protoplasts (left) and tobacco leaves (right). α-Flag IP indicates immunoprecipitation with anti-Flag antibody beads. Experiments in (**a**–**d**) were repeated three times, with similar results. Confocal images and illustrations of OsDAF1-eYFP signal in tetrads of WT (**e**) and *Ossrf8* mutant (**f**) backgrounds. Scale bars, 10 μm. **g** Confocal optical sections and schematic diagrams of tetrads expressing OsSRF8-eYFP protein in the *Osdaf1* background. Five independent T$_1$ lines were imaged, with similar results. Scale bars, 10 μm. The graphs on the right represent the fluorescence values after quantification and the white dashed arrows indicate the direction of fluorescence measurement.

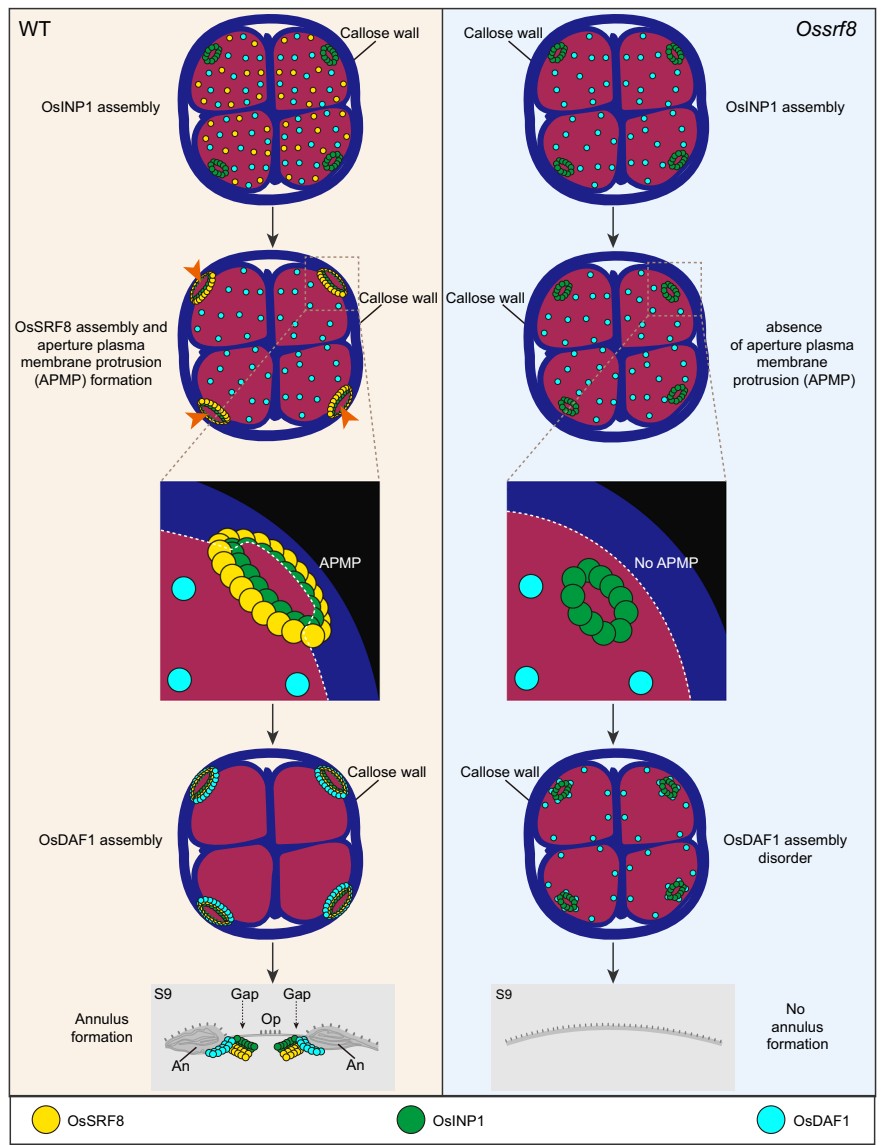

**Fig. 8 | A working model of OsSRF8 during aperture formation.** In the WT background, OsINP1 is polarly aggregated in the pre-aperture region and subsequently recruits OsSRF8, which jointly promotes the formation of APMP. Further, OsINP1 and OsSRF8 are responsible for recruiting OsDAF1, which defines annulus formation. During the S9, OsINP1 and OsSRF8 accumulate below the gap (having no pre-exine/exine region), and OsDAF1 accumulates at the edge of the annulus, which may be involved in regulating the deposition of sporopollenin in the annulus. However, mutation in *OsSRF8* causes absence of APMP and abnormal localization of OsDAF1, eventually forming pollen grains without aperture. The white dotted lines indicate the boundary of the plasma membrane. APMP aperture plasma membrane protrusion, An annulus, Op operculum.

the other from the interactions between the proteins and the PM[33–38]. Previous studies reported that Arabidopsis D6PKL3 and INP1 possess the ability to bind phosphatidylinositol, and that phosphatidylinositol is enriched in the APMP domains, suggesting that the interaction between aperture protein molecules with lipids may be a potential driving force of APMP formation[19]. Combined with the above clues, we speculate that the OsINP1-OsSRF8 protein complex aggregates polarly on the PM, perhaps also due to the attraction of membrane lipids, resulting in changes in the physical or chemical properties of the PM, such as changes in the fluidity of the PM or changes in the composition of the PM, and eventually forcing the PM to form localized protrusions at the APMP sites.

The annulus and operculum identified as the ornament of the aperture are unique structures in Poaceae pollen and their formation follows closely behind the APMP. OsDAF1 acts as a downstream protein, and the OsINP1-OsSRF8 protein complex recruits it into the annulus-forming region, defining the initial annulus. OsDAF1 is a highly conserved protein in the Poaceae that controls the annulus generation and that is localized near the edge of annulus[7]. At the S9 stage, the annulus displays parallel trilamellated structures and can be clearly distinguished (Supplementary Fig. 2e). As the microspores continue to develop, sporopollenin begins to accumulate rapidly at the location of annulus (Supplementary Fig. 2f–h). The Fibrillar-granular layer and Zwischenkörper layer, located under the operculum, begin to form at the S10 stage and are basically completed by the S11 stage (Supplementary Fig. 2f–i). Considering that OsDAF1 is distributed at the edge of the annulus and that the *Osdaf1* mutant lacks the annulus and associated Fibrillar-granular layer[7], we thus speculate that OsDAF1 might be required to regulate the transport of sporopollenin to the annulus region and formation of the Fibrillar-granular layer. However, the detailed molecular mechanisms of OsDAF1 in regulating annulus formation await further studies.

OsSRF8 belongs to the fifth subfamily (LRRV) of the LRR-RLKs proteins[31]. The LRRV subfamily contains nine and twelve members in

Arabidopsis and rice, respectively, which have been shown to play diverse roles in regulating floral organ shape, leaf shape, root hair patterning, plant pathogen response, plant nutritional immunity, genetic incompatibility and cell wall biology[31,39–44]. Our protein structure analysis predicted that the OsSRF8 protein possesses a single transmembrane domain (aa 321-343), dividing the protein into an extracellular domain (ECD, aa 26-320) and an intracellular domain (ICD, aa 344-757), consistent with consensus pattern of the majority of LRR-RLKs proteins[31]. OsDAF1 is also predicted to possess a single transmembrane domain, a canonical N-terminal extracellular domain, and a C-terminal intracellular kinase domain[7]. Notably, we showed that the N-terminal extracellular domain and C-terminal domain of OsSRF8 are responsible for interacting with OsINP1 and OsDAF1, respectively (Figs. 4 and 5a–d). Thus, we speculate that OsINP1 may be located extracellularly, as the Arabidopsis AtINP1[9,19], and that OsINP1 interacts with the N-terminal extracellular domain OsSRF8 outside of the cell to recruit OsSRF8 to the pre-APMP sites. Meanwhile, the C-terminal domain of OsSRF8 interacts with the C-terminal intracellular kinase domain of OsDAF1, recruiting OsDAF1 to the annulus region. However, a previous study reported that OsINP1 interacts with the C-terminal domain OsDAF1[7], raising the intriguing question whether OsINP1 is located extracellular or intracellularly. Two possible scenarios could explain the apparent contradiction: (1) OsINP1 is dually distributed in both the intracellular and extracellular areas (possibly it is a secreted protein), or (2) OsINP1 might contain a potential transmembrane domain and possess both extracellular and intracellular domains. This intriguing question warrants further investigation in future research.

Previous studies have demonstrated that Arabidopsis-derived *STRUBBELIG* (*SUB*) is classified within the SRF family and encodes an atypical kinase, often referred to as a pseudokinase due to the absence of one or more key residues essential for phosphoryl transfer[39,45–47]. Sequence alignment revealed that like SUB, OsSRF8 harbors amino acid alterations in the conserved positions crucial for kinase activity, such as E474M in the C-helix, D556N in the HRD motif, and D574H in the DFG motif (Supplementary Fig. 13). We thus suspect that OsSRF8 is a pseudokinase similar to SUB. Pseudokinase often functions as a scaffold to recruit catalytically active kinases or by altering the conformation of protein chaperones to facilitate signal transduction[48,49]. For example, the pseudokinase CORYNE is a scaffold protein for CLAVATA 2 that transports CLAVATA 2 to the PM and mediates interactions between proteins to regulate stem cell fate[50,51]. The pseudokinase BSKs (BR-signaling kinases) are the substrates for BRI1 kinase and function as scaffolds to activate downstream BR signal transduction[52,53]. The polar accumulation of a pair of pseudokinase PAN2/1 promotes polarized actin accumulation and nuclear polarization in subsidiary mother cells (SMCs) of stomata, resulting in asymmetric division of the SMCs[54–56]. Based on the genetic interactions between OsINP1, OsSRF8, and OsDAF1, it is reasonable to speculate that the pseudokinase OsSRF8 may act as a scaffold protein to mediate interactions with other unknown APMP forming factors to guide APMP generation, meanwhile, OsINP1-OsSRF8 protein complex can guide the localization of OsDAF1 to regulate annulus formation. Overall, the OsINP1-OsSRF8-OsDAF1 molecular module not only deepens our understanding of pollen aperture formation in rice but also in other cereal crops as well.

## Methods

### Plant materials and growth conditions

The *Ossrf8* mutants were isolated from the ethyl methane sulfonate (EMS) mutant library of *Oryza sativa japonica* Ningjing 4. The F$_2$ mapping population was generated from a cross between the *Ossrf8* mutant and the N22 (*Oryza sativa ssp. indica*). The *Osinp1* mutant was generated using the CRISPR/Cas9 system in the Nipponbare background and the deletion of a single base T ultimately leads to premature translation termination at codon 86. The *Osdaf1* mutants were isolated from the EMS mutant library of *Oryza sativa japonica* Ningjing 6 and mutation in single-base G-A results in changes in amino acid, culminating in the formation of aperture without annulus. The allele *daf1-1* mutant obtained by knockout harbors a base A insertion, which ultimately leads to premature translation termination at codon 175. All the rice plants used in this study were grown in the paddy field of Nanjing Agricultural University during natural growing seasons. All the primers used for the mapping experiment are listed in Supplementary Table 13.

### Characterization of the mutant phenotypes

Whole plant architecture, spikelets, and flowers were photographed using a Nikon Z5 digital camera and a Leica stereomicroscope. Mature pollen grains were stained using 1% (w/v) iodine-potassium iodide (I$_2$-KI) solution and observed with a Nikon AZ100 microscope to analyze pollen viability. The surface and interior of mature pollen grains were photographed with SEM and TEM. The observation of embryo sacs was conducted as previously described[57].

### In vitro and in vivo pollen germination assays

In vitro and in vivo pollen germination assays were performed as described[58], with slight modifications. Mature and fresh pollen grains collected just after anthesis were shaken directly onto a liquid germination medium containing 15% (w/v) sucrose, 3 mM calcium nitrate and 40 mg L$^{-1}$ boric acid, followed by incubation in the dark at 30 °C for 15 min. For the vivo pollen-germination assay, spikelets were excised after 2 h of pollination and fixed in Carnoy's solution overnight. The pistils removed from the spikelets were placed in a 70%, 50%, 30% ethanol series, and then washed three times with distilled water. The pistils were then incubated in 1 M NaOH for 30 min at 55 °C to soften them and washed once with distilled water. Finally, the pistils were stained with 0.05% aniline blue (in 0.1 M K$_2$HPO$_4$, pH 8.5) for 1 h in the dark. Pollen tube elongation was observed under a fluorescence microscope (Zeiss Axio Imager 2) in the DAPI channel.

### Sequence and phylogenetic analysis

For the confirmation of OsSRF8 transcripts, we first found that the sequence information predicted by MSU, RAP-DB and NCBI were different to varying degrees, so we employed anther-derived cDNA as a template for amplifying the CDS of OsSRF8, and the sequencing outcomes demonstrated conformity with the NCBI reference (XM_015771292.2). Consequently, in our subsequent experiments, we relied on the transcript information provided by NCBI. The full-length OsSRF8 protein sequence is as follows: MAAAALPRLLLAAAVLCAA-FAPVSGFTDPSDALGLWELYRTLDSPWQLSGWTSQGGDPCGRGGEQRP WHGVLCRDSSIVALNISGLGVGGWLGLELLKFYSLKILDVSFNNIAGEIPRN LPPSVEYLNFAANQFEGSIPPSLPWLHTLKYLNLSHNKLSGIIGDVFVNMES LGTMDLSFNSFSGDLPTSFSSLKNLHHLYLQHNEFTGSVILLADLPLSSLNI ENNSFSGYVPGTFESIPELRIDGNQFQPGFKRASPSFTRSAHSPPTPHPPPSS PPPPMSPPPPAVKENLKHKPEPLKPSLSHSSMYNHNQHRKSHSRVTAAAI ATVTGTAFVLLIVGLVLKSCTYSPKSTANNAKSPPANVEKVPKANEVLYSW NSLMNDCEASSSDVIKPERAMKTKVWAKTSKNFLTAKQFQAVDILAATR NFSKECFIGEGFTGQVYRGDFPGGQLLAIKKINMVDLSLSEQDELIDMLGK MSNLKHPNISALVGYCVEFGHCALLYEYAENGSLDDILFSAATRSRALSW KARMKIALGVAYALEFMHSTCSPPVVHGNIKATNILLDAQLMPYLSHCGL ARLSQFVSAIRTDSEALNSGKGYVAPELTDPATDSIKADIYSFGVILLVLLTG QKAFDSSRRQNEQFLVDWASPHLHNLDSLERITDPRIHASMPPQAISTLG NIILLCIKKSPELRPPMTVITDKLLKLVQSTGLQKTSTTTQHLEVDAQEPSF KTTRPYFEPSFTVSQSATGGCISQR.

Subsequently, the protein sequence respectively was submitted to the SMART, Aramemnon and DeepTMHMM website (http://smart.embl-heidelberg.de, http://aramemnon.botanik.uni-koeln.de, https://dtu.biolib.com/DeepTMHMM) to distinguish signal peptide, transmembrane domain, LRR domain, and kinase domain. For the phylogenetic analysis, the full-length protein sequence of OsSRF8 was used

as a query to search for orthologs in rice and other plant species using the NCBI website. The homologous proteins obtained were aligned using ClustalW, and a phylogenetic tree was constructed basing on the neighbor-joining (NJ), BioNJ, and Subtree-Pruning-Regrafting algorithms in the MEGA 11 software. The statistical significance was evaluated with 1000 bootstrap replicates.

## Plasmid construction and plant transformation

For the functional complementation assay, we cloned the genomic fragments of *OsSRF8* amplified from wild-type rice genomic DNA containing the entire coding sequences and a 2240-bp promoter into the binary vector pCambia1390 by In-Fusion cloning. Using *A. tumefaciens strain* EHA105, the vector was infected into *Ossrf8* heterozygous mutant calli. To construct a fusion complementary vector, the *1390-eYFP/mCherry-3'UTR* backbone was first constructed (*1390-HindIII-Ubi-KpnI-eYFP/mCherry-3'UTR*), in which eYFP/mCherry and 3'UTR (1321 bp base after the *OsSRF8* stop codon) were inserted downstream of the *Ubi* (derived from the maize polyubiquitin gene promoter) promoter by homologous recombination, while a KpnI cleavage site was introduced upstream of the eYFP. The plasmid was then double-digested with HindIII and KpnI, into which the *OsSRF8~pro~:gOsSRF8* fragment (without stop codon) was inserted. To generate the *OsSRF8pro:H2B-EGFP-3'UTR* promoter reporter, the *H2B-EGFP* fusion gene was inserted into the HindIII/BamHI restriction sites downstream of the *OsSRF8* promoter in the pCambia1390 vector. The mature Nipponbare rice seeds were used to induce calli for Agrobacterium-mediated genetic transformation. The construction of the *OsINP1~pro~:gOsINP1-EGFP* and *OsDAF1~pro~:gOsDAF1-eYFP* vector was based on previous reports[7]. All primers used for plasmid construction are listed in Supplementary Table 13.

## Subcellular localization of the OsSRF8 protein

The coding sequence of *OsSRF8* was cloned into the *pCambia1305-35S~pro~-eYFP* vector. The construct was then introduced into *A. tumefaciens strain* EHA105 and transiently co-expressed with the PM marker AtPIP2-mCherry in tobacco leaf cells[59]. Cell plasmolysis was induced by treatment with a 30% sucrose solution for 15 min. The tobacco epidermal cells were observed 48 h after infiltration using a Leica SP8 laser scanning confocal microscope. eYFP fluorescence was imaged at an excitation wavelength of 514 nm and an emission wavelength of 522–555 nm and mCherry fluorescence was imaged at an emission wavelength of 600–700 nm. The coding sequence of *OsSRF8* was cloned into the pAN580-GFP vector and the recombinant plasmid was then transfected into rice protoplasts[60]. After 16 h, the protoplasts were observed under confocal microscopy. All relevant primers used in the subcellular localization assay are listed in Supplementary Table 13.

## Confocal microscopy

To observe changes in the localization of protein fluorescent tags during microspore development, anthers of different stages were dissected and placed into SlowFade™ Diamond Antifade Mountant (Thermo Fisher Scientific), while gently pressing the sample with a coverslip to allow the dye to penetrate. The fluorescence signals of eYFP/EGFP/mCherry protein were captured using a Leica SP8 confocal microscope. More detailed observations of the stained tetrads were performed as previously described[22]. Briefly, anthers from the tetrad stage were dissected, placed in SlowFade™ Diamond Antifade Mountant supplemented with 0.02% calcofluor white and 5 mg ml$^{-1}$ membrane stain Cell Mask Deep Red (Molecular Probes), and then imaged using a confocal microscope. Calcofluor white fluorescence was imaged at an excitation wavelength of 405 nm and an emission wavelength of 424–475 nm, Cell Mask Deep Red fluorescence was imaged at an excitation wavelength of 640 nm and an emission

wavelength of 663–738 nm, eYFP fluorescence was imaged at an excitation wavelength of 514 nm and an emission wavelength of 522 to 555 nm.

## Protein interaction assays

For the yeast two-hybrid (Y2H) assay, the fragments of *OsSRF8* (N), *OsSRF8* (C), *OsINP1, OsDAF1* (N) and *OsDAF1* (C) were cloned into the pGADT7 vector at the EcoRI and BamHI restriction sites; the fragments of *OsSRF8* (N), *OsSRF8* (C), *OsINP1, OsDAF1* (N) and *OsDAF1* (C) were cloned into the pGBKT7 vector at the EcoRI and BamHI restriction sites. OsSRF8 (N) and OsSRF8 (C) respectively represent the N-terminal domain (amino acids 26-320) and C-terminal domain (amino acids 344-757) of the OsSRF8 protein, OsDAF1 (N) and OsDAF1 (C) respectively represent the N-terminal domain (amino acids 24-331) and C-terminal domain (amino acids 355-695) of the OsDAF1 protein. Following the manufacturer's instructions, both the bait and prey plasmids were co-transformed into the yeast competent cells AH109 and grown on selection medium (Clontech).

The bimolecular fluorescence complementation (BiFC) assay was performed as previously described[60]. The sequences of *OsSRF8, OsSRF8* (N) and *OsSRF8* (C) were cloned into the binary vector pSPYNE, and the sequences of *OsINP1* and *OsDAF1(C)* were cloned into the binary vector pSPYCE. The constructs were introduced into *A. tumefaciens strain* EHA105 and the respective combinations of YN and YC were co-infiltrated into 5-week-old *N. benthamiana* leaves. After 48 h, the fluorescence signal was observed under a confocal microscope (Leica SP8).

The firefly luciferase complementation (Luc) assay was performed as previously described[61]. The sequences of *OsSRF8, OsSRF8* (N), *OsSRF8* (C), *OsINP1, OsDAF1* and *OsDAF1* (C) were cloned into PCAMBIA1300-nLuc and PCAMBIA1300-cLuc vectors. The constructs were introduced into *A. tumefaciens strain* EHA105 and the respective combinations of nLuc and cLuc were co-infiltrated into 5-week-old *N. benthamiana* leaves. After 48 h, detached leaves were incubated with 1 mM luciferin for 15 min and the luciferase luminescence signal was recorded using the Tanon 5200 imaging system.

We also performed co-immunoprecipitation assays in *N. benthamiana* leaves and rice protoplasts as previously described[62]. The coding regions of *OsSRF8* (N) and *OsDAF1(C)* were cloned into the binary vector pCAMBIA1300-Flag, and *OsINP1* and *OsSRF8* (C) were cloned into the binary vector pCAMBIA1305-GFP. *OsDAF1* were cloned into the pAN580-Flag vector and *OsSRF8* were cloned into the pAN580-HA vector. Proteins were then extracted in immunoprecipitation (IP) buffer (50 mM Tris-HCl pH 7.5, 100 mM NaCl, 1 mM Na$_2$-EDTA, 5% glycerol [v/v], 0.75% Triton X-100 [v/v], 1 mM PMSF, 50 μM MG132 [MCE], 1× protease inhibitor mixture [Roche]) for 1 h at 4 °C with rotation. The resulting supernatant was combined with 20 μl anti-Flag antibody M2 agarose beads (Sigma) for 2 h at 4 °C with rotation. The beads were washed three times with IP buffer. Proteins were eluted from the beads by boiling in SDS-PAGE sample buffer for 5 min and analyzed by immunoblotting using anti-GFP (Abcam, dilution 1:5000), anti-Flag (MBL M185-7, dilution 1:5000) and anti-HA (MBL M180-7, dilution 1:5000) antibodies. All relevant primers used in the protein interaction assays are listed in Supplementary Table 13.

## Statistics and reproducibility

Numbers of samples or replicates are indicated in figure legends. We used GraphPad Prism 8 for statistical analysis. Chi-square test analysis was processed using Excel 2010.

## Data availability

All data supporting the findings of this study are available within the article, Supplementary Information files or from the corresponding author on request. Source data are provided with this paper.

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

## Acknowledgements

This research was supported by National Key Research and Development Program of China (2022YFF1003503, 2022YFD1200801), the National Nature Science Foundation of China (U2002202, 31991224, 31971909), the Foundation of Biological Breeding Zhongshan Lab (BM2022008-03 and ZSBBL-KY2023-04); Natural Science Foundation of Jiangsu Province (BK20212010), the Key Research and Development Program of Jiangsu Province (BE2021360); the Key Laboratory of Biology, Genetics and Breeding of Japonica Rice in the Mid-lower Yangtze River; and the Jiangsu Collaborative Innovation Center for Modern Crop Production.

## Author contributions

Z.Z., H.W., and J.W. directed the project. K.C. and Q.W. performed most of the experiments. J.G., S.Z., and S.C. performed the rice transformation. X.Y., C.W., S.Y., H.Z., J.L., D.L., and A.J. participated the map-based cloning. X.H., H.Y., X.Z., Y.C., M.Z., K.L., and L.H. performed the TEM and SEM sections. Y.T., X.L., S.L., and L.J. conducted and managed the field work. K.C., Y.B., Z.Z., and H.W. wrote the paper and finalized the paper. Special thanks to Professor Yehui Xiong from the Chinese Agricultural Sciences for his invaluable assistance with protein structure prediction. All the coauthors approved the paper.

## Competing interests

The authors declare no competing interests.
