## [Peer Review File · Nature Communications]

OsSRF8 interacts with OsINP1 and OsDAF1 to regulate pollen aperture formation in riceREVIEWER COMMENTS

Reviewer #1 (Remarks to the Author):

Chen et al. describe a previously uncharacterized gene from rice, OsSRF8, and demonstrate that it plays an essential role in the formation of the single aperture that develops in the pollen of this species. The gene encodes a receptor kinase-like protein that likely lacks kinase activity. Like previously identified rice aperture factors OsINP1 and OsDAF1, the OsSRF8 protein localizes specifically to the aperture domain of the plasma membrane (PM), where it assembles into a ring-like structure. Via several protein-protein interactions assays, the authors demonstrate that OsSRF8 can interact with both OsINP1 and OsDAF1, which have previously been shown to interact with each other. By investigating how the loss of OsINP1 or OsDAF1 affects the ability of OsSRF8 to localize to the aperture PM domain and how the loss of OsSRF8 affects the ability of OsINP1 and OsDAF1 to localize there, the authors conclude that OsINP1 acts upstream of OsSRF8 and is necessary to bring OsSRF8 to the aperture PM domain. They further propose that OsSRF8 acts upstream of OsDAF1 and helps it to localize to the aperture PM domain. Overall, the study is solid, and the findings are interesting.

There are, however, several issues that need to be resolved:

1. The authors propose that the OsSRF8 protein has two transmembrane domains (see Sup. Fig. 6). This is very unconventional for a receptor kinase-like protein, and there is a definite possibility that the proposed topology might not be correct. This prompted me to try and see what topologies would be predicted for this protein by several TM-predicting algorithms, yet I ran into a problem. I retrieved the protein sequence annotated for LOC_Os02g10110 by the MSU Rice Genome annotation but noticed a discrepancy with the sequence shown in the paper. There is a single isoform predicted by MSU, and while most of that protein sequence seems to be identical to the one presented in Sup. Fig. 6, there is a difference: the protein shown in Sup. Fig. 6 is shorter than the MSU one (757 aa vs. 772 aa), with 15 aa missing from the region indicated in brown color in Sup. Fig. 6.

I also searched the RAP-DB annotation: their J-Browser shows two possible isoforms – one that comes from the MSU site, and the other, whose N-terminal part differs significantly from both the MSU one and the one presented by the authors. If the authors base their evidence for the OsSRF8 protein sequence on the analysis of the OsSRF8 anther cDNA that they obtained, this should be clearly described, and it should be indicated where the correct sequence can be retrieved from. It would be also a good idea to correct the publicly available annotations (if they are indeed incorrect), or at least to mention the discrepancy in the Materials and Methods.

2. Coming back to the topology of the protein. When I entered the sequence identical to the one used by the authors into the PROTTER program (which they used to come up with the protein topology shown in Sup. Fig. 6), the output was quite different. Although the program also proposed the existence of two TM domains, it placed the N-terminal and the C-terminal regions outside the cell – a clearly incorrect topology, given that the kinase domain ended up outside the cell. This suggests that this algorithm cannot be reliably used for this particular protein. For TM domain prediction, PROTTER specifically relies on predictions from a single algorithm - Phobius, which indeed predicts two TM domains for OsSRF8. I also tried several more algorithms: DeepTMHMM predicts a topology that is consistent with the typical RLK structures: a signal peptide (1-25 aa), extracellular domain (26-322), TM domain (323-343) and intracellular domain (344-757). In addition, Aramemnon, which uses predictions of several algorithms to come up with the consensus topology, also predicted a single TM domain in the 323-343 region. While multiple programs predicted the existence of a transmembrane region in the ~80-90 aa region, the probability of its existence seemed to be weaker compared to the one in the 323-343 region. I have my doubts about OsSRF8 having the two-TMD structure that the authors so confidently proposed. At the very least, they should describe the complications arising during the analysis and the possibility that their first TMD may not exist.

3. For interactions between OsSRF8 and OsINP1 or OsDAF1, were any other regions tested besides the ones which are described in Fig. 4 and Fig.7. What were the outcomes?

4. The authors describe that localization of OsDAF1-YFP was “completely disrupted and exhibited

random distribution" in the *Ossrf8* mutant (line 290). Yet, the image presented in the Fig. 7f clearly shows that *OsDAF1-YFP* is still enriched at the aperture sites. This result thus requires more careful interpretation. If the area occupied by *OsDAF1-YFP* in the *Ossrf8* mutant appears to be reduced, this should be quantified.

5. Fig. 2b-f: APMPs are hard to see in the small images. Higher-magnification images should be provided. On the other hand, images in Fig. 2l-p are not particularly informative and can be put in supplementary or removed entirely.

6. Fig. 4c and Fig. 7c need labels for columns. Also, YFP signals are very hard to see – will be nearly impossible to see in the publication. The mCherry signals are also only slightly more visible.

7. In Fig. 8, the label for *osDAF1* is impossible to see in the image of the tetrad.

8. Sup. Fig. 2: What is the difference between the images in panels a and b? Protrusions are not very visible, so better images need to be provided. In the legend, "fault" can be replaced with "gap". In d-f, the white dotted lines are barely visible. Also, since "the suspected route" is essentially a speculation, the dotted lines can be removed.

9. Sup. Fig. 4 – why is there such a large difference between the images in the panels a and e? In both cases, WT pollen was used for pollination, but, while in the panel a multiple pollen tubes can be easily seen, in the panel e they are barely visible (and probably there are fewer of them?).

10. Sup. Fig. 9 – In the tobacco leaf epidermal cells, it's very hard to distinguish between the PM and the cytoplasmic signal. This piece of data is not particularly informative and can be removed.

Additional comments:

1. Pollen is normally used as a singular noun ("pollen is", not "pollens are"). If the authors want to use it as a plural noun, they can replace "pollens" with "pollen grains".

2. Line 19 – pollen is not "gametes"; it's a gametophyte.

3. Line 21 and 46 – "male nuclei". It's actually sperm cells, not "male nuclei".

4. Line 27 – "regulatory mechanisms between them" – unclear what it means.

5. Line 28 – "STRUBBELIG" should be capitalized.

6. Line 35 – "to promote plasma membrane polarization". This is not the conclusion that can be drawn from the data. In addition, if *OsINP1* acting upstream of *OsSRF8* is already present at the specific aperture PM domain, it suggests that the membrane is already polarized.

7. Line 43 – "male gametophytes" – here it should be gametes, not gametophytes.

8. Lines 45-46 – "pollen tubes grown from pollen aperture" – are there more than one tube grown from the aperture?

9. Line 47 – "female nuclei" – should be "female gametes".

10. Line 66 – "is regulated in a sporophytic manner". This sentence has to be modified. While all the available data so far show that aperture patterns are indeed regulated in a sporophytic manner, this cannot be automatically deduced from the fact that first signs of apertures are visible at the late tetrad stage. Tetrad is already a gametophytic stage.

11. Line 68 – "pre-exine" – should be "primexine".

12. Line 131 – "these results indicate that absence of APMP.." – a better conclusion here would be that the *Ossrf8* mutation causes the absence of APMP and disrupts formation of pollen aperture.

13. Line 160, "LOC_Os02g10110 represents *OsSRF8*" – again, a better conclusion here would be that LOC_Os02g10110 is the gene whose disruption is responsible for the *Ossrf8* mutant phenotype.

14. Line 205 – "Fig. 2d, I, n" – these figures do not provide any support to the preceding sentence.

15. "mosaicked" – not the right word here. Sandwiched? Present? Located? Found?

16. Lines 228-229 – "an unknown biochemical function protein" – should be "a protein of unknown biochemical function" (the protein itself is not unknown; it's its biochemical function that is unknown)

17. Lines 245-248 – the fact that the phenotype of the double mutant is the same as the

phenotype of the two single mutants does not necessarily allow to conclude that “they act genetically in the same pathway”. The fact that single mutants have identical phenotypes already suggests that they act in the same pathway. The phenotype of the double mutant doesn’t add much beyond this in this case. If you start with inaperturate single mutants, you will likely observe inaperturate double mutants. Complete loss of aperture is already such a severe phenotype that you might not expect to see anything else in the case of the double mutants.

18. Throughout the manuscript, “Mcherry” should be changed to the conventional “mCherry”.

19. Line 280 – specify that it was CRISPR mutagenesis.

20. Lines 283-285 – The result that the Osdaf1 mutants lack starch granules while Osinp1 and Ossf8 have them is confusing. It’s unclear why the starch granules would leak in the absence of the annulus. Also, while it’s OK to speculate about what’s causing this, it’s important to clearly indicate that it’s a speculation.

21. Lines 300-303 – References should be cited here.

22. Line 312 – “plasma” – should be “plasma membrane”.

23. Line 353-354 – “which tends to be transported via the gap to the annulus”. What is the evidence for this? This seems to be a speculation, but it is not identified as such.

24. Line 387 – “stomata aperture” – should be “stomata”.

25. Line 447 – “3’UTR” - Need to indicate which 3’ UTR was used. Was it from OsSRF8? How many nucleotides after the stop codon were used?

26. Line 448 – “Ubi” – need to indicate which ubiquitin promoter and from which organism was used.

27. Lines 453 and 457 – “Calli” should be “calli” – no need to capitalize the first letter.

28. Line 463 – “1305” – is it pCambia1305?

29. Line 470 – “PAN580” – is it pAN580? Reference should be provided.

30. Line 476 – “of different periods” – should be “of different stages”.

31. Line 852 – “pollen fertilities” – it’s viability, not fertility. Also, please indicate (either in this figure legend or in the Materials and Methods) what was used to stain the embryo sacs.

32. Line 914-915 – “Microspore mother cells are distributed within the purple dashed lines”. – The lines are white. Also, while this sentence apparently refers to the panel a, it seems to be present in the description of the panel f.

Reviewer #2 (Remarks to the Author):

In this manuscript, the authors evaluate the molecular mechanism involved in the formation of the pollen aperture in rice, a process that has also been studied in the model plant *Arabidopsis*. This work builds upon the previous study by Zhang et al. 2020 (number 7 in the reference list of the manuscript) by introducing a new player, OsSRF8, which interacts with OsINP1 and OsDAF1. This research provides valuable insights into an essential process for pollen germination in monocots. The study is well addressed through mutagenesis, histology, transgenic plants, and protein-protein interaction experiments. However, my main concern is about the lack of important information in certain key experiments (complementation assay, transgenic plants, and mutant plants, see below) that prevents the evaluation of result consistency. There are also minor and major points that should be addressed before publication.

Line 21. In the sentence, ‘aperture deliver’ must be changed to either ‘aperture, delivering’ or ‘aperture to deliver’.

Line 56 and 57. Change ‘Howerver’ by ‘However’ and ‘Largley’ by ‘Largely’.

Line 70. The author must indicate the reference for the developmental stages (S9, S10...) of the pollen.

In the supplementary Fig. 2, the yellow color representing Gp should be changed to a different color as it is not very clear. I am unable to see the white dotted lines.

Line 138. Although N22 has been described in the material and methods section as *Oryza sativa* ssp. *Indica*, it should also be indicated in the results section for clarity.

Line 151-158, 200-202, 241-247, 252-254, 280-286, 289. The author must provide more data (specifically, numerical data) regarding the complementation assay, transgenic plants, and mutant plants, either in this section or in the Materials and Methods. Specifically, details such as the number of plants evaluated, whether the defective pollen aperture (in the case of mutation

generation), complementation (in the case of functional study), or signal (in the case of YFP reporter) is total or partial, and the generation of the plants (T2, T3, etc.) should be included for clarity.

Line 158. There is no data (numbers) on the sterility of Cr-Ossrf8-1 and Cr-Ossrf8-2.

Line 171. The authors have performed the phylogenetic analysis using the neighbor-joining method. To enhance accuracy, they must also conduct the phylogenetic analysis using maximum parsimony or/and maximum likelihood methods, which provide clade support.

Line 189. To assess whether the signal started to disappear at the vacuolated microspore stage, the authors need to add quantifications to this assessment. How many vacuolated microspores show the signal?

Line 195-198. It is recommended that they should further confirm if OsSRF8 is associated with the plasma membrane or the cell wall in the root. This could be done through plasmolysis experiments in roots of the stable transgenic rice plants expressing eYFP-tagged OsSRF8.

The author used bright field microscopy to observe the pollen grains, but the images are not clear enough in most of the photos, such as in Fig 3A-D, 5F, or 7F. To improve clarity, the authors should use differential interference contrast microscopy.

Lines 235-240, 256. The interaction between OsSRF8 and OsINP1, as well as OsSRF8 and OsDAF1 has been conducted using truncated versions rather than the full-length proteins (except in the case of OsINP1). It is possible that the truncated protein may have a different structure compared to the full-length protein, or that the interaction sites in the truncated version may be buried within the 3D structure of the full-length protein. The authors must also evaluate the interaction experiment, including the use of the full length of the protein, and further explain the use of the truncated version.

Line 403. The authors must provide more details on the generation of Osinp1.

Line 411. The I2-KI solution must be further described.

Reviewer #3 (Remarks to the Author):

The manuscript entitled, "The receptor kinase OsSRF8 interacts with OsINP1 and OsDAF1 to regulate pollen aperture formation in rice", describes phenotypes of ossrf8 mutants (Fig1), cloning of OsSRF8 genes (Fig2), protein localization of OsSRF8(Fig3),OsSRF8 interaction with OsINP1(Fig4),Genetic and local relationship between OsSRF8 and OsINP1(Fig5&6),Genetic and local relationship between OsSRF8 and OsDAF1(Fig7), and their proposed model to explain pollen aperture formation(Fig8).

All the authors' experiments well done. Good data presentation and good writing. I agree that OsSRF8 is a critical key player in rice pollen aperture formation.

However, when reading this manuscript with a few previous papers in this field such as Zhang et al. (2020) , this manuscript does not have a big impact. It is a kind of extension in this field. This impression comes from the authors's explanation in which OsSRF8 is a pseudo kinase-type scaffold and OsINP1 recruit OsSRF8 at the provisional APMP site, indicating no new finding from a biochemical point of view.

Minor comment;

"The receptor kinase OsSRF8" in the title should be changed to "The possible pseudo kinase OsSRF8"

Reviewer #1:

Chen et al. describe a previously uncharacterized gene from rice, OsSRF8, and demonstrate that it plays an essential role in the formation of the single aperture that develops in the pollen of this species. The gene encodes a receptor kinase-like protein that likely lacks kinase activity. Like previously identified rice aperture factors OsINP1 and OsDAF1, the OsSRF8 protein localizes specifically to the aperture domain of the plasma membrane (PM), where it assembles into a ring-like structure. Via several protein-protein interactions assays, the authors demonstrate that OsSRF8 can interact with both OsINP1 and OsDAF1, which have previously been shown to interact with each other. By investigating how the loss of OsINP1 or OsDAF1 affects the ability of OsSRF8 to localize to the aperture PM domain and how the loss of OsSRF8 affects the ability of OsINP1 and OsDAF1 to localize there, the authors conclude that OsINP1 acts upstream of OsSRF8 and is necessary to bring OsSRF8 to the aperture PM domain. They further propose that OsSRF8 acts upstream of OsDAF1 and helps it to localize to the aperture PM domain.

Overall, the study is solid, and the findings are interesting.

Response: We deeply appreciate the reviewer's positive evaluation and helpful suggestions on our manuscript!

There are, however, several issues that need to be resolved:

1. The authors propose that the OsSRF8 protein has two transmembrane domains (see Sup. Fig. 6). This is very unconventional for a receptor kinase-like protein, and there is a definite possibility that the proposed topology might not be correct. This prompted me to try and see what topologies would be predicted for this protein by several TM-predicting algorithms, yet I ran into a problem. I retrieved the protein sequence annotated for LOC_Os02g10110 by the MSU Rice Genome annotation but noticed a discrepancy with the sequence shown in the paper. There is a single isoform predicted by MSU, and while most of that protein sequence seems to be identical to the one presented in Sup. Fig. 6, there is a difference: the protein shown in Sup. Fig. 6 is shorter

than the MSU one (757 aa vs. 772 aa), with 15 aa missing from the region indicated in brown color in Sup. Fig. 6. I also searched the RAP-DB annotation: their J-Browser shows two possible isoforms – one that comes from the MSU site, and the other, whose N-terminal part differs significantly from both the MSU one and the one presented by the authors. If the authors base their evidence for the OsSRF8 protein sequence on the analysis of the OsSRF8 anther cDNA that they obtained, this should be clearly described, and it should be indicated where the correct sequence can be retrieved from. It would be also a good idea to correct the publicly available annotations (if they are indeed incorrect), or at least to mention the discrepancy in the Materials and Methods.

Response: Thanks for the valuable comment. To verify the accuracy of *OsSRF8* transcripts, we examined sequence variations predicted by NCBI, RAP-DB, and MSU. The NCBI results indicated that *OsSRF8* has two transcripts, corresponding to lengths of 2274-bp (XM_015771292.2) and 2310-bp (XM_015771290.2) for the coding sequence (CDS). Additionally, transcripts predicted by the RAP and MSU databases revealed that the CDS of *OsSRF8* encompasses Os02t0194600-00 (1974-bp) and LOC_Os02g10110.1 (2319-bp), respectively. To ensure accuracy, we utilized anther-derived cDNA as a template to amplify the CDS of *OsSRF8*. The sequencing outcomes demonstrated conformity with the NCBI reference (XM_015771292.2). Consequently, in our subsequent experiments, we relied on the transcript information provided by NCBI. Additionally, we have incorporated this sequencing information into the Material and Methods section for comprehensive documentation. (Line 460 – Line 474)

2. Coming back to the topology of the protein. When I entered the sequence identical to the one used by the authors into the PROTTER program (which they used to come up with the protein topology shown in Sup. Fig. 6), the output was quite different. Although the program also proposed the existence of two TM domains, it placed the N-terminal and the C-terminal regions outside the cell – a clearly incorrect topology, given that the kinase domain ended up outside the cell. This suggests that this algorithm cannot be reliably used for this particular protein. For TM domain prediction, PROTTER specifically relies on predictions from a single algorithm - Phobius, which

indeed predicts two TM domains for OsSRF8. I also tried several more algorithms: DeepTMHMM predicts a topology that is consistent with the typical RLK structures: a signal peptide (1-25 aa), extracellular domain (26-322), TM domain (323-343) and intracellular domain (344-757). In addition, Aramemnon, which uses predictions of several algorithms to come up with the consensus topology, also predicted a single TM domain in the 323-343 region. While multiple programs predicted the existence of a transmembrane region in the ~80-90 aa region, the probability of its existence seemed to be weaker compared to the one in the 323-343 region.

I have my doubts about OsSRF8 having the two-TMD structure that the authors so confidently proposed. At the very least, they should describe the complications arising during the analysis and the possibility that their first TMD may not exist.

Response: Thank you very much for your careful reviewing and valuable suggestions on our manuscript. Following your suggestions, we performed a thorough re-analysis of the OsSRF8 protein structure. The protein sequence translated from the correct transcripts of OsSRF8 (XM_015771292.2) was submitted to analysis using the Aramemnon and DeepTMHMM programs, and the results showed that OsSRF8 possesses a single transmembrane domain (see Reply Fig. 1a and b). However, when we utilized the frequently employed PROTTER (<http://wlab.ethz.ch/protter/start/>) and TMHMM 2.0 (<https://services.healthtech.dtu.dk/services/TMHMM-2.0/>) programs for predictions, the outcomes remained in alignment with our previous predictions, indicating the presence of two transmembrane domains (see Reply Fig. 1c and d). Notably, the PROTTER and TMHMM 2.0 programs predict that the C-terminal region of OsSRF8 is situated outside the cell. This finding contradicts the expected localization of the kinase domain, which typically is located intracellularly, implying a potential error in the prediction made by the PROTTER and TMHMM 2.0 programs. Consider OsSRF8 is classified within the LRR-RLK clade V and that reported LRR receptor kinases such as PEPR1/2, GSO1/2, MIK1, BAM1/2, ERECTA, HPCA1, GHR1, TMK1/2, CLV1, BRI1, FLS2, etc., exhibit a single TMD^{1,2,3,4,5,6,7,8,9,10,11}, it is more reasonable to assume that OsSRF8, with a single TMD, aligns with this prevailing pattern.

Based on the above analysis, we tended to agree with your assessment and the prediction results made by the Aramemnon and DeepTMHMM programs. Furthermore, we utilized AlphaFold to depict the predicted protein topology of OsSRF8. The results revealed that OsSRF8 consists of a single transmembrane domain, dividing the protein into an extracellular domain (ECD, aa 26-320), a transmembrane domain (aa 321-343) and an intracellular domain (ICD, aa 344-757). Amino acids 1 to 25 represent the predicted signal peptide (see **Reply Fig. 2** below for Reviewer’s convenience or in Supplementary Fig. 6 in the revised manuscript). These corrections are presented in the revised manuscript.

Reply Fig. 1. The prediction of TMD of OsSRF8 protein from different programs. The images represent the results of the Aramemnon (a), DeepTMHMM (b), PROTTER (c) and TMHMM 2.0 (d) programs for predicting the TMD of OsSRF8 protein, respectively.

Reply Fig. 2. AlphaFold prediction of OsSRF8 topology. OsSRF8 is predicted to have one predicted transmembrane domain that divides the protein into an Extracellular domain (ECD, aa 26-320), a transmembrane domain (aa 321-343) and an intracellular domain (ICD, aa 344-757). Amino acids 1-25 represent the predicted signal peptide.

1. Yu Y, *et al.* ABLs and TMKs are co-receptors for extracellular auxin. *Cell* **186**, 5457-5471.e5417 (2023).
2. Chen L, Cochran AM, Waite JM, Shirasu K, Bemis SM, Torii KU. Direct attenuation of Arabidopsis ERECTA signalling by a pair of U-box E3 ligases. *Nat. Plants* **9**, 112-127 (2023).
3. Roman AO, Jimenez-Sandoval P, Augustin S, Broyart C, Hothorn LA, Santiago J. HSL1 and BAM1/2 impact epidermal cell development by sensing distinct signaling peptides. *Nat. Commun* **13**, 876 (2022).
4. Wu F, *et al.* Hydrogen peroxide sensor HPCA1 is an LRR receptor kinase in Arabidopsis. *Nature* **578**, 577-581 (2020).
5. Doll NM, *et al.* A two-way molecular dialogue between embryo and endosperm is required for seed development. *Science* **367**, 431-435 (2020).
6. Zheng X, *et al.* Danger-Associated Peptides Close Stomata by OST1-

- Independent Activation of Anion Channels in Guard Cells. *Plant Cell* **30**, 1132-1146 (2018).
7. Sierla M, *et al.* The Receptor-like Pseudokinase GHR1 Is Required for Stomatal Closure. *Plant Cell* **30**, 2813-2837 (2018).
 8. Wang T, *et al.* A receptor heteromer mediates the male perception of female attractants in plants. *Nature* **531**, 241-244 (2016).
 9. Sun Y, *et al.* Structural basis for flg22-induced activation of the Arabidopsis FLS2-BAK1 immune complex. *Science* **342**, 624-628 (2013).
 10. Ogawa M, Shinohara H, Sakagami Y, Matsubayashi Y. Arabidopsis CLV3 peptide directly binds CLV1 ectodomain. *Science* **319**, 294 (2008).
 11. He Z, *et al.* Perception of brassinosteroids by the extracellular domain of the receptor kinase BRI1. *Science* **288**, 2360-2363 (2000).

3. For interactions between OsSRF8 and OsINP1 or OsDAF1, were any other regions tested besides the ones which are described in Fig. 4 and Fig.7. What were the outcomes?

Response: Thank you for your constructive suggestions. Following your recommendations, we conducted additional experiments to investigate the interactions between OsSRF8 with OsINP1 or OsDAF1 in different regions. Given the confirmation that OsSRF8 likely possesses a single transmembrane domain, we divided OsSRF8 into two segments: intracellular and extracellular. The interaction between OsSRF8 and OsINP1 and OsDAF1 was initially verified through yeast two-hybrid experiments. The results revealed that only the extracellular domain of OsSRF8 (OsSRF8(N)) interacts with OsINP1 (see reply Fig. 3 below for Reviewer's convenience or in Fig. 4a in the revised manuscript) and that only the intracellular domain of OsSRF8 (OsSRF8(C)) interacts with OsDAF1(C) (see reply Fig. 4 below for Reviewer's convenience or in Fig. 7a in the revised manuscript). The observed interaction between OsSRF8(N) with OsINP1 suggests that OsINP1 is likely extracellularly localized during the tetrad stage, as observed for the Arabidopsis INP1 protein^{1,2}.

Reply Fig. 3. Examination of interaction between OsSRF8 and OsINP1.

a Yeast two-hybrid (Y2H) assay verifies the interaction between OsSRF8 and OsINP1. OsSRF8 (N) and OsSRF8 (C) represent the N-terminal domain (amino acids 26-320) and C-terminal domain (amino acids 344-757) of the OsSRF8 protein, respectively. AD, activating domain; BD, DNA-binding domain; SD, synthetic defined medium.

Reply Fig. 4. Examination of interaction between OsSRF8 and OsDAF1.

a Yeast two-hybrid (Y2H) assay verifies the interaction between OsSRF8 and OsDAF1. OsDAF1(N) and OsDAF1(C) represent the N-terminal domain (amino acids 24-331) and C-terminal domain (amino acids 355-695) of the OsDAF1 protein, respectively. AD, activating domain; BD, DNA-binding domain; SD, synthetic defined medium.

1. Li P, Ben-Menni Schuler S, Reeder SH, Wang R, Suarez Santiago VN, Dobritsa

AA. INP1 involvement in pollen aperture formation is evolutionarily conserved and may require species-specific partners. *J. Exp. Bot.* **69**, 983-996 (2018).

2. Lee BH, *et al.* Arabidopsis Protein Kinase D6PKL3 Is Involved in the Formation of Distinct Plasma Membrane Aperture Domains on the Pollen Surface. *Plant Cell* **30**, 2038-2056 (2018).

4. The authors describe that localization of OsDAF1-YFP was “completely disrupted and exhibited random distribution” in the *Ossrf8* mutant (line 290). Yet, the image presented in the Fig. 7f clearly shows that OsDAF1-YFP is still enriched at the aperture sites. This result thus requires more careful interpretation. If the area occupied by OsDAF1-YFP in the *Ossrf8* mutant appears to be reduced, this should be quantified.

Response: Thank you for pointing out the mistake in our statement. We found that there was a notable shift in the polar localization of OsDAF1-YFP in the *Ossrf8* mutant background. Instead of the OsDAF1-YFP fluorescent signals concentrating in the pre-aperture region in the wild type background, most of the signals exhibited a random distribution on the plasma membrane (PM) in the *Ossrf8* mutant background. To address concerns about potential misrepresentation due to low-quality images, we have also performed new experiment and updated the new data in Fig. 7f in the revised manuscript (also provided in Reply Fig. 5 below for Reviewer’s convenience). To provide a more comprehensive understanding, we quantified the fluorescence signal. The results revealed that the fluorescence values of OsDAF1-YFP in the *Ossrf8* mutant did not align with those observed in the wild-type background (Reply Fig. 5).

Reply Fig. 5. OsSRF8 affects the distribution of OsDAF1 in the pre-aperture region.

a, b Confocal images and illustrations of OsDAF1-eYFP signal in tetrads of WT (**a**) and *Ossrf8* mutant (**b**) backgrounds. Scale bars, 10 µm. **c** Confocal optical sections and schematic diagrams of tetrads expressing OsSRF8-eYFP protein in the *Osdaf1* background. Scale bars, 10 µm. The graphs on the right represent the fluorescence values after quantification and the white dashed arrows indicate the direction of fluorescence measurement.

5. Fig. 2b-f: APMPs are hard to see in the small images. Higher-magnification images should be provided. On the other hand, images in Fig. 2l-p are not particularly informative and can be put in supplementary or removed entirely.

Response: Thank you very much for your comments. Following your suggestions, we have enlarged the APMPs and provided higher-magnification images for the corresponding regions in Fig. 2 in the revised manuscript (also provided in Reply Fig. 6 below for Reviewer's convenience). Additionally, based on your suggestion, images of Fig. 2l-p have been put in supplementary Fig. 5c.

Reply Fig. 6. The observation of APMPs.

a-e Confocal images of tetrads of WT (**a**), the *Oosrf8* mutant rescued by genomic *OsSRF8* sequence (**b**) or *OsSRF8* fused with *eYFP* (**c**), *Cr-Oosrf8-1* (**d**) and *Cr-Oosrf8-2* (**e**). APMPs are marked with the dashed square. Scale bars, 10 μm. **f-j** The images of the enlarged APMP regions corresponding to the dashed areas in a-e, respectively. The red dotted lines indicate the boundary of the plasma membrane and the blue dotted lines indicate the outer boundary of the callose wall of microspore at the tetrad stage. Scale bars, 3 μm.

6. Fig. 4c and Fig. 7c need labels for columns. Also, YFP signals are very hard to see – will be nearly impossible to see in the publication. The mCherry signals are also only slightly more visible.

Response: Thank you for your valuable suggestion. In accordance with your suggestions, we have repeated the experiment, and the latest results clearly showing the fluorescence of YFP signals are presented in the revised Fig. 4c and Fig. 7c (also see Reply Figs. 7 and 8 below for Reviewer's convenience).

Reply Fig. 7. Interaction between OsSRF8 and OsINP1 detected using Bimolecular fluorescence complementation (BiFC) assay.

BiFC assay verifies the interaction between OsSRF8 and OsINP1. OsINP1 and OsSRF8 or OsSRF8(N) respectively were fused to the C- or N-terminal parts of YFP and co-transformed into tobacco leaves. Scale bars, 50 μ m.

Reply Fig. 8. Interaction between OsSRF8 and OsDAF1 detected using Bimolecular fluorescence complementation (BiFC) assay.

BiFC assay verifies the interaction between OsSRF8 and OsDAF1. OsDAF1 or OsDAF1(C) and OsSRF8 or OsSRF8(C) respectively were fused to the C- or N-terminal parts of YFP and co-transformed into tobacco leaves. Scale bars, 50 μ m.

7. In Fig. 8, the label for osDAF1 is impossible to see in the image of the tetrad.

Response: Thank you very much for your comments. Based on your suggestion, we have modified the label colors, and they are now distinctly visible. Furthermore, we also revised the model diagram (Fig. 8) in the revised manuscript, emphasizing on the role of *OsSRF8* for formation the aperture plasma membrane protrusion (also provided in Reply Fig. 9 below for Reviewer's convenience).

Reply Fig. 9. A working model of OsSRF8 during aperture formation. In the WT background, OsINP1 is polarly aggregated in the pre-aperture region and subsequently recruits OsSRF8, which jointly promotes the formation of APMP. Further, OsINP1 and OsSRF8 are responsible for recruiting OsDAF1, which defines annulus formation. During the S9, OsINP1 and OsSRF8 accumulate below the gap (having no pre-exine/exine region), and OsDAF1 accumulates at the edge of the annulus, which may be involved in regulating the deposition of sporopollenin in the annulus. However, mutation in *OsSRF8* causes absence of APMP and abnormal localization of OsDAF1, eventually forming pollen grains without aperture. The white dotted lines indicate the boundary of the plasma membrane. APMP, aperture plasma membrane protrusion; An, annulus; Op, operculum.

8. Sup. Fig. 2: What is the difference between the images in panels a and b? Protrusions are not very visible, so better images need to be provided. In the legend, “fault” can be replaced with “gap”. In d-f, the white dotted lines are barely visible. Also, since “the suspected route” is essentially a speculation, the dotted lines can be removed.

Response: Thank you for your comments. To enhance the visibility of these differences between the Supplementary Fig. 2a and 2b, we have magnified the regions of aperture in Supplementary Fig. 2 in the revised manuscript (also provided in Reply Fig. 10 below for Reviewer’s convenience). The results now clearly illustrate these distinctions. Additionally, in the legend, we have replaced the term "fault" with "gap". Finally, we've removed the white dotted lines that represent the suspected route in Supplementary Fig. 2f-h in the revised manuscript (also provided in Reply Fig. 10 below for Reviewer’s convenience).

Reply Fig. 10. Developmental process of rice pollen aperture formation. a-d Plasma membrane polarization and the formation of APMP. APMP forms on the surface of tetrad microspores. APMP is marked with the dashed square. (b) and (d) represent enlarged views of the dashed square, respectively. The red dotted lines indicate the boundary of the plasma membrane and the blue dotted lines indicate the boundary of the callose wall. Scale bars, 10 μm in (a, c), 2 μm in (b, d). e-k The decoration of aperture. When the tetrad microspores are separated, the annulus and operculum begin to develop. The Gap (no exine deposition) between the annulus and the operculum has

become clearly visible (e). The sporopollenin aggregates at the annulus, and the continuously enlarged vacuole concentrates the cytoplasm while compressing the Fibrillar-granular layer (f-h). Fibrillar-granular layer is continuously compressed and the Zwischenkörper layer is formed (i). The aperture is formed completely (j). The pollen tube protrudes from the aperture (k). An, annulus; Op, operculum; F, Fibrillar-granular layer; Z, Zwischenkörper layer. Scale bars, 2 μm in (e-j), 4 μm in (k).

9. Sup. Fig. 4 – why is there such a large difference between the images in the panels a and e? In both cases, WT pollen was used for pollination, but, while in the panel a multiple pollen tubes can be easily seen, in the panel e they are barely visible (and probably there are fewer of them?).

Response: Thank you for your comments. The original Sup. Fig. 4a represents self-pollination (WT x WT), while Sup. Fig. 4e depicts artificial pollination (*Ossrf8* x WT). During the artificial pollination process, it is possible that an insufficient amount of wild-type's pollen landed on the stigma of the *Ossrf8* mutant, leading to the relatively fewer pollen tubes germinating on the stigma of the mutant. To validate this hypothesis, we conducted a new pollination experiment with the artificial saturation pollination (enough pollen for pollination). The updated results demonstrated that when *Ossrf8* mutant is saturately pollinated with wild-type's pollen, the pollen tubes germinated normally, similar to the spontaneous pollen tubes in the wild-type in Supplementary Fig. 4 in the revised manuscript (also provided in Reply Fig. 11 below for Reviewer's convenience). This outcome further corroborates our hypothesis.

Reply Fig. 11 Observation of in vivo germination of *Ossrf8* mutant and wild-type pollen grains. **a, b** WT self-pollinated. **c, d** *Ossrf8* mutant self-pollinated. **e, f** *Ossrf8* mutant as the maternal and the WT as the paternal for saturation pollination experiments. **g, h** WT as the maternal and the *Ossrf8* mutant as the paternal for saturation pollination experiments. The orange arrows indicate pollen tubes that grow into the pistils. Scale bars, 200 μ m.

10. Sup. Fig. 9 – In the tobacco leaf epidermal cells, it's very hard to distinguish between the PM and the cytoplasmic signal. This piece of data is not particularly informative and can be removed.

Response: Thank you very much for your constructive comments. Reviewer #2 also raised similar comments. Based on the helpful comments from you and Reviewer #2, we have made the following significant changes: 1) We performed plasmolysis experiments using tobacco epidermal cells and the result effectively confirmed the localization of OsSRF8 on the PM (see Supplementary Fig. 10 in the revised manuscript or in Reply Fig. 17 below for Reviewer's convenience). 2) Based on your suggestion, we have removed the images of tobacco epidermal cells transiently expressing eYFP and OsSRF8-eYFP.

Additional comments:

1. Pollen is normally used as a singular noun (“pollen is”, not “pollens are”). If the authors want to use it as a plural noun, they can replace “pollens” with “pollen grains”.

Response: Thanks for the valuable point. Based on your suggestion, we have revised the description of pollen in the full manuscript.

2. Line 19 – pollen is not “gametes”; it's a gametophyte.

Response: Thanks for the comment. Based on your suggestion, we have changed “The mature male gametes (pollens) of higher plants” into “The mature male gametes of higher plants”. (Lines 20)

3. Line 21 and 46 – “male nuclei”. It's actually sperm cells, not “male nuclei”.

Response: Yes, we have modified it based on your suggestion. (Lines 22 and 48)

4. Line 27 – “regulatory mechanisms between them” – unclear what it means.

Response: Thanks for the comment. We have made the following changes to a description: “Previous studies have identified OsINP1 and OsDAF1 as two essential regulators of APMP and pollen aperture formation in rice. However, the precise

molecular mechanisms of their linkage remain elusive”. (Lines 26-28)

5. Line 28 – “STRUBBELIG” should be capitalized.

Response: Yes, we have modified it based on your suggestion.

6. Line 35 – “to promote plasma membrane polarization”. This is not the conclusion that can be drawn from the data. In addition, if OsINP1 acting upstream of OsSRF8 is already present at the specific aperture PM domain, it suggests that the membrane is already polarized.

Response: Thanks for the comment. We concluded that OsINP1 and OsSRF8 act together to promote APMP formation based on the following observations: 1. The localization of OsINP1 to the pre-aperture region is not affected in the *Ossrf8* mutant background (Fig. 5e). On the contrary, the localization of OsSRF8 to the pre-aperture region is disrupted in the *Osinp1* mutant background (Fig. 5f). 2. The *Ossrf8* mutant exhibits a no aperture phenotype exactly as the *Osinp1* mutant (Fig. 5b). These observations suggest that OsINP1 acts upstream of OsSRF8 and that OsINP1 by itself is not sufficient to promote APMP formation. Instead, APMP formation requires the co-action of OsINP1 and OsSRF8. Therefore, we have made the following changes to a description: “At the tetrad stage, OsSRF8 protein is recruited by OsINP1 to the pre-aperture region through direct protein-protein interaction. Subsequently, they work together to promote the formation of APMPs”. (Lines 35-37)

7. Line 43 – “male gametophytes” – here it should be gametes, not gametophytes.

Response: Thanks for the suggestion. We have revised it. (Line 45)

8. Lines 45-46 - “pollen tubes grown from pollen aperture” – are there more than one tube grown from the aperture?

Response: Thanks for the valuable point. Rice has only one pollen aperture, therefore, only one pollen tube grows from the pollen aperture. Based on your suggestion, we have changed “pollen tubes grown from pollen aperture” into “pollen tube grows from

pollen aperture”. (Lines 47-48)

9. Line 47 – “female nuclei” – should be “female gametes”.

Response: Thanks for the suggestion. We have revised it. (Line 49)

10. Line 66 – “is regulated in a sporophytic manner”. This sentence has to be modified. While all the available data so far show that aperture patterns are indeed regulated in a sporophytic manner, this cannot be automatically deduced from the fact that first signs of apertures are visible at the late tetrad stage. Tetrad is already a gametophytic stage.

Response: Thank the reviewer for careful reviewing our manuscript. We fully agree with reviewer’s suggestion. While aperture patterns are indeed governed in a sporophytic manner, we acknowledge that the initial signs of apertures become evident during the late tetrad stage. Consequently, based on your suggestion, we have revised the description as following: "Like exine formation, the first sign of aperture in rice is readily visible at the late tetrad stage, indicating that the aperture pattern is determined during microsporogenesis". (Lines 65-67)

11. Line 68 – “pre-exine” – should be “primexine”.

Response: As suggested, we have changed “pre-exine” into “primexine”.

12. Line 131 – “these results indicate that absence of APMP...” – a better conclusion here would be that the *Ossrf8* mutation causes the absence of APMP and disrupts formation of pollen aperture.

Response: Thanks for the valuable point. Based on your suggestion, we have changed “these results indicate that absence of APMP...” into “these results indicate that the *Ossrf8* mutation causes the absence of APMP and disrupts formation of pollen aperture...”. (Lines 132-133)

13. Line 160, “LOC_Os02g10110 represents OsSRF8” – again, a better conclusion here would be that LOC_Os02g10110 is the gene whose disruption is responsible for

the *Ossrf8* mutant phenotype.

Response: Thanks for the suggestion. We have changed “*LOC_Os02g10110* represents OsSRF8” into “*LOC_Os02g10110* is the gene whose disruption is responsible for the *Ossrf8* mutant phenotype”. (Lines 162-163)

14. Line 205 – “Fig. 2d, I, n” – these figures do not provide any support to the preceding sentence.

Response: Thanks for pointing out this. Based on your suggestion, we have added the description of these figures in the revised manuscript as following: “Pollen aperture was restored in all positive transgenic plants, indicating that the transgene is functional (Figs. 2d, i, n and Supplementary Table 5)”. (Lines 201-203)

15. “mosaicked” – not the right word here. Sandwiched? Present? Located? Found?

Response: Thanks for the suggestion. We have changed “mosaicked” into “sandwiched”. (Line 215)

16. Lines 228-229 – “an unknown biochemical function protein” – should be “a protein of unknown biochemical function” (the protein itself is not unknown; it’s its biochemical function that is unknown).

Response: Thanks for the valuable point. Based on your suggestion, we have changed “an unknown biochemical function protein” into “encoding a protein of unknown biochemical function”. (Lines 220-221)

17. Lines 245-248 – the fact that the phenotype of the double mutant is the same as the phenotype of the two single mutants does not necessarily allow to conclude that “they act genetically in the same pathway”. The fact that single mutants have identical phenotypes already suggests that they act in the same pathway. The phenotype of the double mutant doesn’t add much beyond this in this case. If you start with inaperturate single mutants, you will likely observe inaperturate double mutants. Complete loss of aperture is already such a severe phenotype that you might not expect to see anything

else in the case of the double mutants.

Response: Thank you for your valuable insight. In accordance with your suggestions, we have revised the description as follows: “The identical phenotypes observed in the single mutants of *Osinp1* and *Ossrf8* suggest that they operate within the same genetic pathway to regulate aperture pattern in pollen surface. To further explore potential synergistic interactions between *OsSRF8* and *OsINPI*, we generated the *Ossrf8/Osinp1* double mutant. As expected, the phenotype of the double mutant was identical to that of the single mutant, verifying that *OsSRF8* and *OsINPI* act in the same genetic pathway to regulate pollen aperture formation (Fig. 5c and Supplementary Fig. 11a-f)”. (Lines 239-245)

18. Throughout the manuscript, “Mcherry” should be changed to the conventional “mCherry”.

Response: As suggested, we have revised them throughout the manuscript.

19. Line 280 – specify that it was CRISPR mutagenesis.

Response: As suggested, we have revised it as following: “At the same time, *Ossrf8/Osinp1/Osdaf1-1* triple mutants were obtained using the CRISPR/Cas9 technology to knock out *OsDAF1* in the background of heterozygous *Ossrf8/Osinp1* mutants (Supplementary Fig. 11g, k, l, n and Supplementary Table 10)”. (Lines 280-283)

20. Lines 283-285 – The result that the *Osdaf1* mutants lack starch granules while *Osinp1* and *Ossrf8* have them is confusing. It’s unclear why the starch granules would leak in the absence of the annulus. Also, while it’s OK to speculate about what’s causing this, it’s important to clearly indicate that it’s a speculation.

Response: Thank you for your comments. Based on your suggestion, we have revised this description as follows: “Iodine-potassium iodide (I₂-KI) staining showed that most of the *Osdaf1* mutant pollen were aborted. We speculated that the abortion of the *Osdaf1* mutant pollen is likely caused by leakage of starch granules due to the absence of the

annulus.” (Lines 284-286)

21. Lines 300-303 – References should be cited here.

Response: As suggested, we have cited references in these sentences. (Line 306)

22. Line 312 – “plasma” – should be “plasma membrane”.

Response: As suggested, we have changed “plasma” into “plasma membrane”. (Line 315)

23. Line 353-354 – “which tends to be transported via the gap to the annulus”. What is the evidence for this? This seems to be a speculation, but it is not identified as such.

Response: Thank you for your comments. Indeed, at present, we lack direct evidence to demonstrate that sporopollenin is transported via the gap to the annulus. Based on your suggestion, we have revised this description as follows: “Considering that OsDAF1 is distributed at the edge of the annulus and that the *Osdaf1* mutant lacks the annulus and associated Fibrillar-granular layer¹, we thus speculate that OsDAF1 might be required to regulate the transport of sporopollenin to the annulus region and formation of the Fibrillar-granular layer”. (Lines 357-361)

1. Zhang X, *et al.* Rice pollen aperture formation is regulated by the interplay between OsINP1 and OsDAF1. *Nat. Plants* **6**, 394-403 (2020).

24. Line 387 – “stomata aperture” – should be “stomata”.

Response: As suggested, we have changed “stomata aperture” into “stomata”. (Line 402)

25. Line 447 – “3’UTR” - Need to indicate which 3’ UTR was used. Was it from OsSRF8? How many nucleotides after the stop codon were used?

Response: Thanks for the valuable point. The 3’UTR (1321 bp) is derived from OsSRF8. (Line 491)

26. Line 448 – “Ubi” – need to indicate which ubiquitin promoter and from which organism was used.

Response: Thanks for the point. Ubi is derived from the maize polyubiquitin gene promoter. (Lines 493-494)

27. Lines 453 and 457 – “Calli” should be “calli” – no need to capitalize the first letter.

Response: As suggested, we have revised it. (Lines 490 and 500)

28. Line 463 – “1305” – is it pCambia1305?

Response: Yes, 1305 is pCambia1305. Based on your suggestion, we have revised it. (Line 507)

29. Line 470 – “PAN580” – is it pAN580? Reference should be provided.

Response: Thanks for the suggestion. We have revised it and provided the corresponding reference. (Lines 515-516)

30. Line 476 – “of different periods” – should be “of different stages”.

Response: As suggested, we have revised it. (Line 521)

31. Line 852 – “pollen fertilities” – it’s viability, not fertility. Also, please indicate (either in this figure legend or in the Materials and Methods) what was used to stain the embryo sacs.

Response: As suggested, we have revised it as following: “Pollen grains were stained using 1% I₂-KI solution”. The observation of embryo sacs was conducted as previously described¹. (Line 895)

1. Yu Y, *et al.* Hybrid Sterility in Rice (*Oryza sativa* L.) Involves the Tetratricopeptide Repeat Domain Containing Protein. *Genetics* **203**, 1439-1451 (2016).

32. Line 914-915 – “Microspore mother cells are distributed within the purple dashed

lines”. – The lines are white. Also, while this sentence apparently refers to the panel a, it seems to be present in the description of the panel f.

Response: Thank you for your comments. We have revised it as following: “The fluorescence signal of H2B-EGFP is firstly observed in the microspore mother cell (MMCs) and MMCs are demarcated by the white dashed lines (a)”. (Lines 939-940)

Reviewer #2:

In this manuscript, the authors evaluate the molecular mechanism involved in the formation of the pollen aperture in rice, a process that has also been studied in the model plant *Arabidopsis*. This work builds upon the previous study by Zhang et al. 2020 (number 7 in the reference list of the manuscript) by introducing a new player, OsSRF8, which interacts with OsINP1 and OsDAF1. This research provides valuable insights into an essential process for pollen germination in monocots.

The study is well addressed through mutagenesis, histology, transgenic plants, and protein-protein interaction experiments. However, my main concern is about the lack of important information in certain key experiments (complementation assay, transgenic plants, and mutant plants, see below) that prevents the evaluation of result consistency. There are also minor and major points that should be addressed before publication.

Response: We deeply appreciate the reviewer's positive evaluation and helpful suggestion on our manuscript.

Line 21. In the sentence, 'aperture deliver' must be changed to either 'aperture, delivering' or 'aperture to deliver'.

Response: Thank you for your comments. Based on your suggestion, we have revised this sentence to make it accurate.

Line 56 and 57. Change 'Howerver' by 'However' and 'Largley' by 'Largely'.

Response: Thanks for the suggestion. Based on your suggestion, we have made modifications to these words.

Line 70. The author must indicate the reference for the developmental stages (S9, S10...) of the pollen.

Response: As suggested, we have revised it.

In the supplementary Fig. 2, the yellow color representing Gp should be changed to a different color as it is not very clear. I am unable to see the white dotted lines.

Response: Thank you for your comments. We have revised this picture to make it clearer (supplementary Fig. 2 or Reply Figure 10). Based on the Reviewer's suggestion, the white dotted line has been removed.

Line 138. Although N22 has been described in the material and methods section as *Oryza sativa* ssp. Indica, it should also be indicated in the results section for clarity.

Response: Thanks for the suggestion. We have modified it.

Line 151-158, 200-202, 241-247, 252-254, 280-286, 289. The author must provide more data (specifically, numerical data) regarding the complementation assay, transgenic plants, and mutant plants, either in this section or in the Materials and Methods. Specifically, details such as the number of plants evaluated, whether the defective pollen aperture (in the case of mutation generation), complementation (in the case of functional study), or signal (in the case of YFP reporter) is total or partial, and the generation of the plants (T2, T3, etc.) should be included for clarity.

Response: Thank you for your constructive suggestion. We have added the data to Supplementary Tables 1-12.

Line 158. There is no data (numbers) on the sterility of Cr-Ossrf8-1 and Cr-Ossrf8-2.

Response: Thank you for your comments. Based on your suggestion, we have added the data of seed setting rate for *Cr-Ossrf8-1* and *Cr-Ossrf8-2* (see Supplementary Fig. 5 in the revised manuscript or in Reply Fig. 12 below for Reviewer's convenience).

Reply Fig. 12. Phenotypes of the *OsSRF8* knockout and complemented plants. a-d Morphological phenotypes of WT, *Ossrf8*, *Com-OsSRF8/Ossrf8*, *Cr-Ossrf8-1* and *Cr-Ossrf8-2* plants (a), flowers (b), mature pollen grains (c) and mature panicles (d). Scale bars, 20 cm in (a), 2 mm in (b), 25 μ m in (c) and 5 cm in (d). e statistics of seed setting rate of WT, *Ossrf8*, *Com-OsSRF8/Ossrf8*, *Cr-Ossrf8-1* and *Cr-Ossrf8-2* mature panicles.

Line 171. The authors have performed the phylogenetic analysis using the neighbor-joining method. To enhance accuracy, they must also conduct the phylogenetic analysis using maximum parsimony or/and maximum likelihood methods, which provide clade support.

Response: Thank you for your comment. Based on your suggestion, we conducted the phylogenetic analysis using the three methods: neighbor-joining method (Reply Figure 13 below for Reviewer's convenience), maximum parsimony (Reply Figure 14 below for Reviewer's convenience) and maximum likelihood method (Reply Figure 15 below for Reviewer's convenience). The results obtained from all three methods were

consistent with our previously provided results. This reaffirms that the *OsSRF8* gene is indeed unique to the Poaceae family. Consequently, we have selected to present the result obtained by the neighbor-joining method in the revised manuscript (see Supplementary Fig. 7 in the revised manuscript or in Reply Figure 13 below for Reviewer's convenience).

Reply Fig. 13. Evolutionary tree analysis of the OsSRF8 protein based on the neighbor-joining method.

OsSRF8 protein belongs to the class of proteins of STRUBBELIG-RECEPTOR FAMILY (SRF) and is highly conserved within grass species. By homology analysis of the OsSRF8 protein sequence with NCBI BLAST, we obtained several protein sequences that score relatively high in different species. In the Poaceae, the similarity

of OsSRF8 is relatively high, all around 70%, while in other species, the similarity is less than 50%. The phylogenetic analysis was performed with MEGA 11 version, using the neighbor-joining (NJ) algorithm. The numbers on the left represent branch length.

Reply Fig. 14. Evolutionary tree analysis of the OsSRF8 protein based on the maximum parsimony method.

OsSRF8 protein belongs to the class of proteins of STRUBBELIG-RECEPTOR FAMILY (SRF) and is highly conserved within grass species. By homology analysis of the OsSRF8 protein sequence with NCBI BLAST, we obtained several protein sequences that score relatively high in different species. In the Poaceae, the similarity

of OsSRF8 is relatively high, all around 70%, while in other species, the similarity is less than 50%. The phylogenetic analysis was performed with MEGA 11 version, applying Neighbor-Join and BioNJ algorithms.

Reply Fig. 15. Evolutionary tree analysis of the OsSRF8 protein based on the maximum likelihood method.

OsSRF8 protein belongs to the class of proteins of STRUBBELIG-RECEPTOR FAMILY (SRF) and is highly conserved within grass species. By homology analysis of the OsSRF8 protein sequence with NCBI BLAST, we obtained several protein sequences that score relatively high in different species. In the Poaceae, the similarity of OsSRF8 is relatively high, all around 70%, while in other species, the similarity is

less than 50%. The phylogenetic analysis was performed with MEGA 11 version, using the Subtree-Pruning-Regrafting (SPR) algorithm.

Line 189. To assess whether the signal started to disappear at the vacuolated microspore stage, the authors need to add quantifications to this assessment. How many vacuolated microspores show the signal?

Response: Thank you for your comment. Based on your suggestion, we investigated the nuclear EGFP fluorescence signal at the vacuolated microspore stage. The results revealed that out of 57 microspores, only 14 retained the fluorescence signal (Supplementary Table 4). Furthermore, we have duly adjusted the corresponding text as following: “However, upon investigation of the microspore fluorescence signal at the vacuolated microspore stage, it was observed that only 14 out of 57 microspores retained fluorescence. This suggests that the signal started to disappear at the vacuolated microspore stage (Supplementary Fig. 9 and Supplementary Table 4 in the revised manuscript or Reply Fig. 16 below for Reviewer’s convenience)”. (Lines 188-192)

Reply Fig. 16. *OsSRF8* is expressed during aperture development.

a-f Images of different periods of microspore development expressing the transcriptional fusion construct *OsSRF8pr:H2B-EGFP*. The fluorescence signal of H2B-EGFP is firstly observed in the microspore mother cell (MMCs) distributed within

the white dashed lines (a), dyad-stage (b), tetrad stage (c), early-stage young free microspore (d) and free microspore (e). However, the signal started to disappear at the vacuolated microspore stage (14 non-fluorescent / total 57 microspores) (f). White arrows indicate H2B-EGFP fluorescence signal and the fluorescence signal located at the aperture is derived from autofluorescence. CW, calcofluor white; MMCs, microspore mother cells; Ms, microspores. Scale bars, 20 μ m.

Line 195-198. It is recommended that they should further confirm if OsSRF8 is associated with the plasma membrane or the cell wall in the root. This could be done through plasmolysis experiments in roots of the stable transgenic rice plants expressing eYFP-tagged OsSRF8.

Response: Thank you for your constructive comments. Reviewer #1 also raised some similar comments. Based on the helpful comments from you and Reviewer#1's suggestion, we performed plasmolysis experiments using tobacco epidermal cells. The results showed that before plasmolysis, the plasma membrane and cell wall overlapped with each other and could not be clearly distinguished (see Supplementary Fig. 10 in the revised manuscript or in Reply Figure 17 below for Reviewer's convenience). After the plasmolysis experiment, it showed that OsSRF8 protein was localized on the plasma membrane, rather than on the cell wall (see Supplementary Fig. 10 in the revised manuscript or in Reply Fig. 17 below for Reviewer's convenience). In addition, plasmolysis experiments were also performed in roots of the stable transgenic rice plants expressing eYFP-tagged OsSRF8. Unfortunately, we did not observe the fluorescence of OsSRF8 protein, except for the fluorescence of the cell wall. We speculate that the expression of OsSRF8 protein in the root tip is too low for detection.

Reply Fig. 17. OsSRF8 is a membrane protein

a-o Co-expression of the PM marker AtPIP2-mCherry and the full-length OsSRF8 proteins without Plasmolysis (**a-e**) and with plasmolysis (**f-o**) in tobacco epidermal cells. Scale bars, 10 μm in (**a-j**), 20 μm in (**k-o**).

The author used bright field microscopy to observe the pollen grains, but the images are not clear enough in most of the photos, such as in Fig 3A-D, 5F, or 7F. To improve clarity, the authors should use differential interference contrast microscopy.

Response: Thank you for your comments. Based on your suggestion, we conducted the experiments again, and obtained sufficiently clear photos. Consequently, we have adjusted the images in Fig. 3a-d, Fig. 5f and Fig. 7f in the revised manuscript for better clarity (or in Reply Figures 18-20 below for Reviewer's convenience).

Reply Fig. 18. Laser confocal images showing aperture development. Confocal imaging and illustrations of microspores expressing *pOsSRF8:gOsSRF8-eYFP* at S7 (a), S8a (b), S8b (c and d). White arrows indicate the future aperture sites on the tetrad. Scale bars, 20 μm in (a-c), 10 μm in d.

Reply Fig. 19. OsSRF8 affects the distribution of OsDAF1 in the pre-aperture region.

Confocal images and illustrations of *OsDAF1-eYFP* signal in the *Ossrf8* mutant background. Scale bars, 10 μm . The graphs on the right represent the fluorescence values after quantification and the white dashed arrows indicate the direction of

fluorescence measurement.

Reply Fig. 20. OsINP1 affects the distribution of OsSRF8 in the pre-aperture region.

Confocal optical sections and schematic diagrams of tetrads expressing OsSRF8-eYFP protein in the *Osinp1* background. Scale bars, 10 μm. The graphs on the right represent the fluorescence values after quantification and the white dashed arrows indicate the direction of fluorescence measurement.

Lines 235-240, 256. The interaction between OsSRF8 and OsINP1, as well as OsSRF8 and OsDAF1 has been conducted using truncated versions rather than the full-length proteins (except in the case of OsINP1). It is possible that the truncated protein may have a different structure compared to the full-length protein, or that the interaction sites in the truncated version may be buried within the 3D structure of the full-length protein. The authors must also evaluate the interaction experiment, including the use of the full length of the protein, and further explain the use of the truncated version.

Response: Thank you for your comments. Based on your suggestion, we used the full-length OsSRF8 protein to perform interaction experiments with OsINP1 and OsDAF1, respectively. The results from both the BiFC and LUC assays demonstrated that the full-length OsSRF8 protein could interact with OsINP1 and OsDAF1. In addition, the truncated protein OsSRF8(N) also could interact with OsINP1, and OsSRF8(C) could interact with OsDAF1(C) (Fig. 4b-c and Fig. 7b-c in the revised manuscript, or in Reply Figures 21a-b and 22a-b below for Reviewer's convenience). These results once again support that OsSRF8 protein indeed could interact with OsINP1 and OsDAF1. In order to further verify these interactions, we employed the CO-IP method to determine the interaction between full-length protein OsSRF8/OsINP1 and OsSRF8/OsDAF1 in rice

protoplasts (Fig. 7d in the revised manuscript, or in Reply Figs. 21c-d and 22c-d below for Reviewer's convenience). The results showed that the full-length OsSRF8 protein could interact with OsDAF1 protein in rice protoplasts. In addition, the truncated protein OsSRF8(C) also could interact with OsDAF1(C) (Fig. 7d in the revised manuscript or in Reply Fig. 22d below for Reviewer's convenience). However, we encountered difficulties in confirming the interaction between the full-length OsSRF8 and OsINP1 (Reply Fig. 21c-d). Subsequently, we utilized the truncated protein OsSRF8(N) to investigate its interaction with OsINP1 in the CO-IP assay, and the results indicated that truncated protein OsSRF8(N) indeed exhibited interaction with OsINP1 (Fig. 4d in the revised manuscript or in Reply Figures 21e below for Reviewer's convenience). Our analysis suggests that the extracellular domain of OsSRF8, containing the LRR motif, may not be sufficiently stable to reliably in vitro capture the interaction between the full-length OsSRF8 protein and OsINP1 (Reply Figures 21c-d).

Reply Fig. 21. Examination of the interaction between OsSRF8 and OsINP1.

a Interaction between OsINP1 and OsSRF8 detected using Bimolecular fluorescence complementation (BiFC) assay. OsINP1 and OsSRF8/OsSRF8(N) respectively were fused to the C or N-terminal parts of YFP and co-transformed into tobacco leaves. Scale bars, 50 μm . **b** Split-luciferase assay testing the interaction between OsSRF8 and OsINP1 in the tobacco (*Nicotiana benthamiana*) leaves. **c-e** Co-immunoprecipitation experiments in the rice protoplasts (c and d) or tobacco leaves (e). α -Flag IP indicates immunoprecipitation with anti-Flag antibody beads.

Reply Fig. 22. Examination of the interaction between OsSRF8 and OsDAF1.

a Interaction between OsDAF1 and OsSRF8 detected using bimolecular fluorescence complementation (BiFC) assay. OsDAF1/OsDAF1(C) and OsSRF8/OsSRF8(C) respectively were fused to the C or N-terminal parts of YFP and co-transformed into tobacco leaves. Scale bars, 50 μm. **b** Split-luciferase assay testing the interaction between OsSRF8 and OsINP1 in the tobacco (*Nicotiana benthamiana*) leaves. **c-e** Co-immunoprecipitation experiments in the rice protoplasts (c and d) or tobacco leaves (e). α-Flag IP indicates immunoprecipitation with anti-Flag antibody beads.

Line 403. The authors must provide more details on the generation of Osinp1.

Response: Thank you for your comments. Based on your suggestion, we have revised

this description as following: “The *Osinp1* mutant was created employing the CRISPR/Cas9 system in the Nipponbare background. The deletion of a single base T ultimately results in premature translation termination at codon 86” (Lines 417-419).

Line 411. The I₂-KI solution must be further described.

Response: Thanks for the comment. Following the suggestion, we have further described I₂-KI solution as “Mature pollen grains were stained using 1% (w/v) iodine-potassium iodide (I₂-KI) solution” (Lines 431-432).

Reviewer #3:

The manuscript entitled, "The receptor kinase OsSRF8 interacts with OsINP1 and OsDAF1 to regulate pollen aperture formation in rice", describes phenotypes of *ossrf8* mutants (Fig1), cloning of OsSRF8 genes (Fig2), protein localization of OsSRF8(Fig3), OsSRF8 interaction with OsINP1(Fig4), Genetic and local relationship between OsSRF8 and OsINP1(Fig5&6), Genetic and local relationship between OsSRF8 and OsDAF1(Fig7), and their proposed model to explain pollen aperture formation (Fig8). All the authors' experiments well done. Good data presentation and good writing. I agree that OsSRF8 is a critical key player in rice pollen aperture formation.

Response: We highly appreciate the Reviewer's time and efforts in reviewing our manuscript.

However, when reading this manuscript with a few previous papers in this field such as Zhang et al. (2020), this manuscript does not have a big impact. It is a kind of extension in this field. This impression comes from the authors's explanation in which OsSRF8 is a pseudo kinase-type scaffold and OsINP1 recruit OsSRF8 at the provisional APMP site, indicating no new finding from a biochemical point of view.

Response: Thank you for your comment. We respectfully disagree with the Reviewer's viewpoint. Pollen aperture formation is not only important for plant reproduction, but also an interesting biological question. So far, only two genes, OsINP1 and OsDAF1, have been reported to be involved in the formation of pollen aperture in rice, and there remain significant gaps in our comprehension of this important biological process. For instance, although previous studies have shown that *OsINP1* is essential for the formation of aperture plasma membrane protrusions (AMAP), it encodes a protein of unknown biochemical function¹. It remains essentially unclear exactly how OsINP1 promotes AMAP formation. Additionally, although previous studies have shown that the *OsDAF1* gene is required for annulus formation, its action mechanisms also remain essentially unknown. In this study, we report the isolation and identification of the gene *OsSRF8*, which encodes a predicted receptor kinase required for pollen aperture

formation in rice. The novelty and significance of our work include: 1. We found that OsSRF8 is essential for APMP and pollen aperture formation, just as OsINP1; 2. We found that OsSRF8 is an essential partner of OsINP1 for promoting APMP formation, a critical step for aperture formation. This conclusion is based on the following observations: 1) localization of OsINP1 to the pre-aperture region is not affected in the *Ossrf8* mutant background (Fig. 5e); 2) the localization of OsSRF8 to the pre-aperture region is disrupted in the *Osinp1* mutant background (Fig. 5f); 3) the *Ossrf8* mutant exhibits a no aperture phenotype exactly as the *Osinp1* mutant (Fig, 1d). These observations suggest that *OsINP1* acts upstream of *OsSRF8* and that *OsINP1* by itself is not sufficient to promote APMP formation. Instead, APMP formation requires the co-action of *OsINP1* and *OsSRF8*. We thus deduce that through the interaction, OsINP1 not only helps to recruit OsSRF8 to the pre-APMP region, but also may activate OsSRF8 to activate downstream events, leading to APMP formation (Reply Fig. 23a). Additionally, we found that OsSRF8 also physically interacts with OsDAF1 and helps to recruits OsDAF1 to the APMP site to co-regulate formation of the annulus (Reply Figs. 23b and c). Thus, our findings not only identified an essential regulator of pollen aperture formation, but also provides significant new insights into the molecular mechanism of OsINP1 and OsDAF1 in promoting APMP and annulus formation.

Reply Fig. 23. The indispensability of OsSRF8 in pollen aperture formation.

a Both OsSRF8 and OsINP1 are required to form APMPs, which define the initiation site for aperture development. Through direct interaction, OsINP1 not only recruits OsSRF8 to the pre-APMP region, but also may collaboratively regulate APMP formation. **b-c** Subsequently, OsDAF1 is recruited by the OsINP1-OsSRF8 protein complex to APMP region to determine the annulus space. An, annulus; Op, operculum; Gap, having no pre-exine/exine region.

1. Zhang X, *et al.* Rice pollen aperture formation is regulated by the interplay between

OsINP1 and OsDAF1. *Nat. Plants* **6**, 394-403 (2020).

Minor comment;

"The receptor kinase OsSRF8" in the title should be changed to "The possible pseudo kinase OsSRF8"

Response: Thank you for your comment. Based on your suggestion, we have revised it.

REVIEWERS' COMMENTS

Reviewer #1 (Remarks to the Author):

The manuscript has been significantly improved. I only have several minor comments:

1. Title: If you want to keep the information about the pseudokinase nature of OsSRF8 in the title, then I would suggest to change it to "The putative receptor pseudokinase...".
2. Line 20 – pollen grains are gametophytes, not gametes. Gametes are sperm cells. You cannot thus say that gametes have apertures.
3. Line 28, "of their linkage" – unclear what's meant here. What linkage are you talking about?
4. Line 34, "diffused" – should be "diffusely distributed".
5. Line 49, "female gamete" – should be "egg cell" (since central cell is also a female gamete).
6. Line 49, "polar nucleus" – should be "central cell".
7. Line 154, "or plus eYFP" – should be "with or without eYFP".
8. Lines 245-246, "verifying that OsSRF8 and OsINP1 act in the same genetic pathway...". This statement should be removed. (I suggest putting the period just before this phrase). While it was certainly worth checking the phenotype of the double mutant, the result doesn't verify that the genes act in the same genetic pathway (it doesn't contradict this hypothesis, but it also cannot provide clear evidence for it). The evidence comes from other experiments.
9. Line 389, "STRUBBELIG" – please indicate that you talk about the Arabidopsis protein and indicate that it's the founding member of the STRUBBELIG receptor family.
10. Line 393, "Gly-Met", "Asp-Asn", "Asp-His" – please make it clear that in these pairs of amino acids the first ones indicate amino acids found at the corresponding positions in active kinases, while the second one indicates amino acids found in OsSRF8 and SUB. At the moment, it's not very clear.
11. Line 494, "3' UTR" – indicate that it's OsSTR8 3'UTR. Also, it would be a good place to transfer here the detailed information about 3' UTR ("1321 bp after...") from line 492.
12. Line 537 (Protein interaction assays) – it would be worth indicating here the exact fragments (listing amino acid positions) that were used for protein interaction assays. It's helpful to have such information in Methods, even if it was already mentioned in Results. Also, for OsDAF1, I don't think it was provided in Results, making it even more important to mention this here.
13. It is fairly conventional for Y2H experiments to provide plasmid transformation controls (i.e. yeast growth on SD -Trp/-Leu plates). Fig. 4A and Fig. 7A lack such controls.
14. Legend for Sup. Fig. 1, line 865, "keep the pollen grains airtight" – unclear what is meant by "airtight" here.

Reviewer #2 (Remarks to the Author):

In the revised manuscript, the majority of the questions have been addressed satisfactorily. However, some minor points still require the authors' attention.

Line 82: It is recommended that the authors include the name of the orthologous gene. In Supplemental Table 1, clarification is needed for the terms 'Positive' and 'Negative' to enhance clarity.

Line 169-171. While the authors have conducted three phylogenetic analyses, there are still gaps in the data, and the confidence of the inferred relationships cannot be assessed. Support values at nodes, such as bootstrap support in the phylogenetic trees based on Maximum Likelihood or Maximum Parsimony, are absent. As a result, the validity of the concluding statement is compromised (lines 173-175). While this is a relatively minor point in the manuscript, it is essential for it to be accurately addressed or rephrased.

Supplemental table 5 (Line 17-19). The transgene should be indicated in the legend of the table. Supplementary Tables 2, 6, 9 and 10. The data in these tables need further clarification from the authors. I presume that the numbers represent phenotypes of plants, indicating normal or defective pollen apertures, etc. It seems that all the pollen grains exhibit the same phenotype. Additionally, it is recommended that the authors include the phenotype of the wildtype for a comprehensive analysis.

Reviewer #3 (Remarks to the Author):

I think that the revised manuscript has been improved much according to the editor and reviewers comments. In addition, I appreciate this work very much. This is a very valuable new information in this field. Furthermore, I know well that this is a very important biological event to form specific biological hole in pollen and should be solved scientifically.

However, I am afraid that the impact of their finding is not enough for this journal.

In the reply by authors,

Response: Thank you for your comment. We respectfully disagree with the Reviewer's viewpoint. - removed - For instance, although previous studies have shown that OsINP1 is essential for the formation of aperture plasma membrane protrusions (AMAP), it encodes a protein of unknown biochemical function¹. It remains essentially unclear exactly how OsINP1 promotes AMAP formation. Additionally, although previous studies have shown that the OsDAF1 gene is required for annulus formation, its action mechanisms also remain essentially unknown.

This paper may introduce the third important gene to form annulus formation although its action mechanisms remain essentially unknown again.

Do you think that it has enough novelty for the publication to Nature Plants? I think we need some data to imply biochemical function of one of those three genes for the publication. Or physiological or biological roles of one of those proteins at least.

I prefer that this type of finding can be published in journals such as The Plant Cell or Molecular Plant, although I appreciate this work a lot and like it very much.

Reviewer #1:

The manuscript has been significantly improved. I only have several minor comments:

Response: We appreciate the reviewer's positive evaluation and helpful suggestions on our manuscript!

1. Title: If you want to keep the information about the pseudokinase nature of OsSRF8 in the title, then I would suggest to change it to "The putative receptor pseudokinase...".

Response: Thank you very much for your comment. Based on your suggestion and editorial requests, we have changed the title to "OsSRF8 interacts with OsINP1 and OsDAF1 to regulate pollen aperture formation in rice".

2. Line 20 – pollen grains are gametophytes, not gametes. Gametes are sperm cells. You cannot thus say that gametes have apertures.

Response: Thanks for the suggestions. We have modified it based on your suggestion. (Line 20)

3. Line 28, "of their linkage" – unclear what's meant here. What linkage are you talking about?

Response: Thanks for the comment. Based on your suggestion and editorial requests, we have revised the description as following: "Previous studies identified OsINP1 and OsDAF1 as essential regulators of APMP and pollen aperture formation in rice, but their precise molecular mechanisms remain unclear". (Lines 26-28)

4. Line 34, "diffused" – should be "diffusely distributed".

Response: Thanks for the comment. Following your suggestion, we have changed "diffused" to "diffusely distributed" in the revised manuscript. (Lines 32-33)

5. Line 49, “female gamete” – should be “egg cell” (since central cell is also a female gamete).

Response: Thanks for the comment. Following your suggestion, we have changed “female gamete” to “egg cell” in the revised manuscript. (Lines 44-45)

6. Line 49, “polar nucleus” – should be “central cell”.

Response: Thanks for the comment. Following your suggestion, we have changed “polar nucleus” to “central cell” in the revised manuscript. (Line 45)

7. Line 154, “or plus eYFP” – should be “with or without eYFP”.

Response: Thanks for the comment. Following your suggestion, we have changed “or plus eYFP” to “with or without eYFP”. (Line 148)

8. Lines 245-246, “verifying that OsSRF8 and OsINP1 act in the same genetic pathway....”. This statement should be removed. (I suggest putting the period just before this phrase). While it was certainly worth checking the phenotype of the double mutant, the result doesn’t verify that the genes act in the same genetic pathway (it doesn’t contradict this hypothesis, but it also cannot provide clear evidence for it). The evidence comes from other experiments.

Response: Thank you very much for your comments. Following your suggestions, we have removed this statement in the revised manuscript.

9. Line 389, “STRUBBELIG” – please indicate that you talk about the Arabidopsis protein and indicate that it’s the founding member of the STRUBBELIG receptor family.

Response: Thanks for the comment. Following your suggestion, we have revised the description: “Previous studies have demonstrated that Arabidopsis-derived *STRUBBELIG* (*SUB*) is classified within the SRF family and encodes an atypical

kinase, often referred to as a pseudokinase due to the absence of one or more key residues essential for phosphoryl transfer”. (Lines 381-384)

10. Line 393, “Gly-Met”, “Asp-Asn”, “Asp-His” – please make it clear that in these pairs of amino acids the first ones indicate amino acids found at the corresponding positions in active kinases, while the second one indicates amino acids found in OsSRF8 and SUB. At the moment, it’s not very clear.

Response: Thanks for the comment. Following your suggestion, we have revised the description: “Sequence alignment revealed that like SUB, OsSRF8 harbors amino acid alterations in the conserved positions crucial for kinase activity, such as E474M in the C-helix, D556N in the HRD motif and D574H in the DFG motif (Supplementary Fig. 13)”. (Lines 384-387)

11. Line 494, “3’ UTR” – indicate that it’s OsSTR8 3’UTR. Also, it would be a good place to transfer here the detailed information about 3’ UTR (“1321 bp after....”) from line 492.

Response: Thanks for the comment. Following your suggestion, we have changed “3’ UTR” to “3’UTR (1321 bp base after the *OsSRF8* stop codon)” in the revised manuscript. (Line 488)

12. Line 537 (Protein interaction assays) – it would be worth indicating here the exact fragments (listing amino acid positions) that were used for protein interaction assays. It’s helpful to have such information in Methods, even if it was already mentioned in Results. Also, for OsDAF1, I don’t think it was provided in Results, making it even more important to mention this here.

Response: Thanks for pointing it out. Based on your suggestion, we have provided the exact fragments information in Methods. (Lines 536-540)

13. It is fairly conventional for Y2H experiments to provide plasmid transformation

controls (i.e. yeast growth on SD -Trp/-Leu plates). Fig. 4A and Fig. 7A lack such controls.

Response: Thanks for pointing it out. Based on your suggestion, we have provided the plasmid transformation controls in Fig. 4A and Fig. 7A.

14. Legend for Sup. Fig. 1, line 865, “keep the pollen grains airtight” – unclear what is meant by “airtight” here.

Response: Thanks for the comment. Following your suggestion, we have revised the description: “The Fibrillar-granular layer and Zwischenkörper layer bridge the annulus and the operculum, effectively sealing the pollen grains”. (Legend for Sup. Fig. 1, Lines 12-14)

Reviewer #2:

In the revised manuscript, the majority of the questions have been addressed satisfactorily. However, some minor points still require the authors' attention.

Response: We deeply appreciate the reviewer's positive evaluation and helpful suggestions on our manuscript!

Line 82: It is recommended that the authors include the name of the orthologous gene.

Response: Thanks for pointing it out. Based on your suggestion, we have added the name of the orthologous gene. (Line 78)

In Supplemental Table 1, clarification is needed for the terms 'Positive' and 'Negative' to enhance clarity.

Response: As suggestion, we have added a detailed description to enhance clarity between "Positive" and "Negative" in Supplemental Table 1.

Line 169-171. While the authors have conducted three phylogenetic analyses, there are still gaps in the data, and the confidence of the inferred relationships cannot be assessed. Support values at nodes, such as bootstrap support in the phylogenetic trees based on Maximum Likelihood or Maximum Parsimony, are absent. As a result, the validity of the concluding statement is compromised (lines 173-175). While this is a relatively minor point in the manuscript, it is essential for it to be accurately addressed or rephrased.

Response: Thanks for pointing it out. Based on your suggestion, we have provided the bootstrap support in the phylogenetic trees.

Supplemental table 5 (Line 17-19). The transgene should be indicated in the legend of the table.

Response: Thanks for pointing it out. We have indicated the transgene in the legend of the table. (Lines 17-19)

Supplementary Tables 2, 6, 9 and 10. The data in these tables need further clarification from the authors. I presume that the numbers represent phenotypes of plants, indicating normal or defective pollen apertures, etc. It seems that all the pollen grains exhibit the same phenotype. Additionally, it is recommended that the authors include the phenotype of the wildtype for a comprehensive analysis.

Response: Thanks for your suggestion. Based on your suggestion, we have provided a detailed description of these data in Supplementary Tables 2, 6, 9 and 10. Phenotype of the wildtype was included for a comprehensive analysis.

Reviewer #3:

I think that the revised manuscript has been improved much according to the editor and reviewers comments. In addition, I appreciate this work very much. This is a very valuable new information in this field. Furthermore, I know well that this is a very important biological event to form specific biological hole in pollen and should be solved scientifically.

However, I am afraid that the impact of their finding is not enough for this journal. In the reply by authors, response: Thank you for your comment. We respectfully disagree with the Reviewer's viewpoint. - removed - For instance, although previous studies have shown that OsINP1 is essential for the formation of aperture plasma membrane protrusions (AMAP), it encodes a protein of unknown biochemical function¹. It remains essentially unclear exactly how OsINP1 promotes AMAP formation.

Additionally, although previous studies have shown that the OsDAF1 gene is required for annulus formation, its action mechanisms also remain essentially unknown.

This paper may introduce the third important gene to form annulus formation although its action mechanisms remain essentially unknown again.

Do you think that it has enough novelty for the publication to Nature Plants? I think we need some data to imply biochemical function of one of those three genes for the publication. Or physiological or biological roles of one of those proteins at least.

I prefer that this type of finding can be published in journals such as The Plant Cell or Molecular Plant, although I appreciate this work a lot and like it very much.

Response: Thank you for your comment. In this study, we not only identified a novel key regulator of pollen aperture formation in rice, OsSRF8, but also showed that OsSRF8 protein is recruited by OsINP1 to the pre-aperture region through direct protein-protein interaction to promote plasma membrane polarization and formation of APMP. This finding adds new insight into the action mechanism of OsINP1, i.e., OsINP1 promotes APMP formation by recruiting OsSRF8 to the pre-APMP site to trigger APMP formation. Additionally, we showed that the OsINP-OsSRF8 protein complex further recruits OsDAF1 to the APMP site through physical interaction to co-

regulate formation of annulus. These new findings bring a major step forward in our understanding of the cellular and molecular mechanisms controlling pollen aperture formation in cereal species, and thus we believe that this study merits publication in a broad impact journal like *Nature Communications*. Future studies will aim to elucidate the detailed biochemical/biophysical mechanism of these protein complexes in regulating aperture formation.